# Spectral Ghost in Representation Learning: from Component Analysis to Self-Supervised Learning

## Abstract

Self-supervised learning (SSL) has improved empirical performance by unleashing the power of unlabeled data for practical applications. Specifically, SSL extracts the representation from massive unlabeled data, which will be transferred to a plenty of down streaming tasks with limited data. The significant improvement on diverse applications of representation learning has attracted increasing attention, resulting in a variety of dramatically different self-supervised learning objectives for representation extraction, with an assortment of learning procedures, but the lack of a clear and unified understanding. Such an absence hampers the ongoing development of representation learning, leaving a theoretical understanding missing, principles for efficient algorithm design unclear, and the use of representation learning methods in practice unjustified. The urgency for a unified framework is further motivated by the rapid growth in representation learning methods. In this paper, we are therefore compelled to develop a principled foundation of representation learning. We first theoretically investigate the sufficiency of the representation from a spectral representation view, which reveals the spectral essence of the existing successful SSL algorithms and paves the path to a unified framework for understanding and analysis. Such a framework work also inspires the development of more efficient and easy-to-use representation learning algorithms with principled way in real-world applications.

## 1 Introduction

Representation learning has demonstrated tremendous success across a wide variety of machine learning tasks—ranging from classical regression and classification (Chen et al., 2020; Grill et al., 2020), to object detection (He et al., 2020), causal inference (Wang et al., 2022; Sun et al., 2024), and reinforcement learning (Ren et al., 2022b; Zhang et al., 2022)—and across multiple modalities—such as text, images, videos, audios and time series (Meng et al., 2021; Giorgi et al., 2020; Goyal et al., 2021; Schiappa et al., 2023; Liu et al., 2022; Wickstrøm et al., 2022). Such methods have improved empirical performance by unleashing the power of unlabeled data for practical applications in a *self-supervised* way. Specifically, these algorithms formulate a training task based on unlabeled data—*e.g.*, next-token prediction (Radford et al., 2018; 2019; Brown et al., 2020), masked token prediction (Mikolov et al., 2013; Devlin et al., 2018; He et al., 2022), noise contrastive classification (Chen et al., 2020; Oord et al., 2018), or denoising (Vincent et al., 2008)—and learn a data representation by completing such a pre-defined task. The diverse application of representation learning has attracted increasing attention in the machine learning community, resulting in a variety of learning objectives, such as contrastive (Oord et al., 2018; Radford et al., 2021) vs. non-contrastive (Zbontar et al., 2021; Garrido et al., 2022) losses, stationary (Chen et al., 2020; He et al., 2020) vs. non-stationary negative sampling (Grill et al., 2020), and so on.

Given the large number of of distinct paradigms proposed for representation learning algorithms with empirical successes, there have been a few attempts to clarify the connections (Balestriero & LeCun, 2022; Balestriero et al., 2023), however, a clear understanding of desired properties of the representation is still absent. Meanwhile, even though they follow the same *self-supervised* spirit, a common framework for characterizing the properties of the existing SSL algorithms, and analyzing the relationships among these methods is long

overdue. These gaps create an unnecessarily high barrier to representation learning research and algorithm design, both from the theoretical and empirical perspectives.

Concretely, despite the empirical successes achieved by representation from SSL, there are essential research questions have yet to be resolved, *i.e.*,

- *What representation is sufficient and effective for variety of downstream tasks?*

- *How can such a representation learned in an efficient and scalable way?*

which have never been answered explicitly. Here, by "sufficient" we mean that the downstream tasks can be completed by composing functions only upon these representations, instead of from the original data. This requires the representations to capture enough information while ignoring unnecessary details. By "effective", we mean that the composed function upon representation for downstream tasks should be light-weight, rather than a large deep models. By "efficient" and "scalable", we mean the optimization for the representation learning should be with less computational and memory cost, and can be easily scaled up. The unlaid theoretical foundation casts a shadow over the empirical successes SSL representation learning.

On the other hand, due to disparate origins and the variety of algorithmic realizations, current representation learning algorithms use distinct components and exhibit dramatically different behaviors. Without a unified understanding, it is not clear whether the existing methods achieve the desired properties, and difficult to analyze these algorithms in the same framework for fair comparison.

The absence of a unified framework for representation learning also leads to unnecessary barriers for pragmatic researchers. One must exhaustively probe these algorithms to fully grasp the dizzying set of choices in each ad hoc recipe in an isolated way, which not only makes the application of such algorithms extremely complicated, but also potentially compromises empirical performance due to suboptimal design of algorithm components. Such unnecessary complications also limit the use of representation learning methods in broader applications.

These urgent questions are in dire need for investigation in SSL to answer the foundation question with a unified view.

**Contribution.** In this paper, we reveal the sufficiency of representation from spectral view, under which we develop a unified framework that empowers theoretical researchers to analyze existing algorithms, develop novel algorithms, and to extend the range of viable applications for representation learning. Particularly, our contribution lies in following aspects.

- We provide the *spectral view* for characterizing the desired properties to the sufficient representation in Section 2. Moreover, we also reveal the sufficient representation purely upon unlabeled data, which paves the way for efficient spectral representation learning.

- The spectral view induces *a unified framework* for representation learning that reveals relationships between existing SSL methods, provides theoretical justification and rigorous categorization in Section 3, 4, 5, and 6, and builds connections to classic component analysis in Section 8.

- The unified framework also inspires a variety of novel scalable representation learning algorithms in Section 3.2, and 5.2 for different spectral representations.

- Finally, we exploit spectral representations for controllable synthesis in generative models, reinforcement learning, and causal inference, which extends their applicability and effectiveness in Section 9.

## 2 Sufficiency of Spectral Representation

To reveal the desired properties for the sufficient representation, we start with prediction tasks for estimating $\mathbb{E}[y|x]$ from data, in which one needs to learn a function $f(\cdot): \mathcal{X} \to \mathcal{Y}$, where $\mathcal{Y}$ can be $\mathbb{R}^k$ for regression, 1-of-$k$ for classification, and more complex structured space. We denote

$$\mathbb{P}(y|x) = \langle \phi(x), \mu(y) \rangle, \tag{1}$$

as the singular value decomposition (SVD) of the conditional operator $\mathbb{P}(y|x)$, where $\phi(\cdot): \mathcal{X} \to \mathbb{R}^d$ and $\mu(\cdot): \mathcal{Y} \to \mathbb{R}^d$. We emphasize the SVD of any $\mathbb{P}(y|x)$ always exists, but can potentially be infinite dimension $d$. For notation simplicity, we denote $\langle \phi(x), \mu(y) \rangle = \phi(x)^\top \mu(y)$ for finite $d$-dimensional representation, and $\langle \phi(x), \mu(y) \rangle = \int \phi_\omega(x) \mu_\omega(y) p(\omega) d\omega$ for finite $d$-dimensional representation with $p(\omega)$ as some measure over indexing variable $\omega$. Then, we have

$$\mathbb{E}[y|x] = \int y\mathbb{P}(y|x)\,dy = \left\langle \phi(x), \underbrace{\int y\mu(y)\,dy}_{w} \right\rangle \tag{2}$$

which means the optimal mean square solution can be *sufficiently* represented by $\phi(x)$ in a *linear form*, therefore, satisfying our requirement.

However, the $\phi(x)$ is task-dependent as a factorization of $\mathbb{P}(y|x)$, therefore, it is only sufficient for a particular regression task upon $\mathbb{P}(y|x)$. For a different task induced by a different $\tilde{\mathbb{P}}(y|x) = \langle \tilde{\phi}(x), \tilde{\mu}(y) \rangle$, we may need different $\tilde{\phi}(x)$, whose corresponding subspace is different from $\phi(x)$.

A straightforward idea is seeking the *complete space* composed by all the subspaces $\phi(x)$ for all $\mathbb{P}(y|x)$, denoted as $\psi(x)$. With the complete $\psi(x)$ provided, it is enough to predict arbitrary dependent $y$ in any down stream tasks. Specifically, for arbitrary task with a corresponding $\mathbb{P}(y|x)$, there exists one subset of $\psi(x)$ that covers the singular subspace of $\mathbb{P}(y|x)$, and therefore, is sufficient for representing $\mathbb{E}[y|x]$ linearly. It is intractable to enumerate all the possible $\mathbb{P}(y|x)$ for all random variable $y$ to recover the complete space for sufficient representation $\psi(s)$. We consider a $\nu(z)$, which has the same rank with $\psi(x)$, and forms $\mathbb{P}(z|x) := \langle \psi(x), \nu(z) \rangle$, as the *core set*. Although $\nu(z)$ and the core set is still conceptual and unknown, we consider

$$\mathbb{P}(x'|x) = \int \mathbb{P}(x'|z)\mathbb{P}(z|x)\,dz = \mathbb{P}(x') \left\langle \psi(x'), \underbrace{\left( \int \frac{\nu(z)\nu(z)^\top}{\mathbb{P}(z)}\,dz \right)}_{A} \psi(x) \right\rangle, \tag{3}$$

$$= \mathbb{P}(x') \langle \psi(x'), A\psi(x) \rangle, \tag{4}$$

with the second equation comes from Bayes' rule, *i.e.*, $\mathbb{P}(x'|z) = \frac{\mathbb{P}(z|x')\mathbb{P}(x')}{\mathbb{P}(z)}$. This implies we can extract the sufficient representation $\psi(x)$ from (3) purely from unlabeled data, rather than exploiting prediction tasks with labeled data.

In (3), as $\nu(z)$ sharing the same rank to $\psi(x)$, $A \in \mathbb{R}^{d \times d}$ is a full-rank positive-definite matrix, which can be factorized as $A = B^\top B$ with $B \in \mathbb{R}^{d \times d}$, which leads to

$$\mathbb{P}(x'|x) = \mathbb{P}(x') \left\langle \underbrace{B\psi(x')}_{\varphi(x')}, B\psi(x) \right\rangle = \mathbb{P}(x') \langle \varphi(x'), \varphi(x) \rangle \Rightarrow \frac{\mathbb{P}(x, x')}{\mathbb{P}(x)\mathbb{P}(x')} = \langle \varphi(x'), \varphi(x) \rangle. \tag{5}$$

Since $A$ is full-rank, so as $B$. Therefore, the linear space constructed by $\psi(x)$ and $\varphi(x)$ are the same. Therefore, we can also represent $\mathbb{E}[y|x]$ linearly with $\varphi(x)$, *i.e.*, $\mathbb{E}[y|x] = \langle \varphi(x), w \rangle$, through the spectral decomposition of Pointwise Mutual Information (PMI), implied by (5).

As we have clarified the requirement for "sufficient and effective" representation from the spectral perspective, the immediate followup question is whether the current representations learned from the existing SSL methods satisfy the condition, which has been answered affirmatively in following sections.

**Remark (Task-Dependent Spectral Representation):** We have completed the sufficiency in representation through spectral view for $\mathcal{X}$, which seeks for complete space from the extra neighborhood information $\mathbb{P}(x'|x)$. Conceptually, the extracted spectral representation $\varphi(x)$ from $\mathbb{P}(x'|x)$ will be sufficient for arbitrary down stream tasks. However, the general sufficient representation includes all the spectral representations from arbitrary $\mathbb{P}(y|x)$, which might be not necessary for some scenarios where the downstream tasks are already known. For example, in natural language processing, the downstream tasks can be recast as next-token

prediction (Radford et al., 2019), and in multi-modality problems, the downstream tasks can be recast as retrieval (Radford et al., 2021), one can extract the corresponding task-dependent spectral representation, instead of the generic sufficient spectral representation for statistical, memory and computational efficiency. Particularly, in multi-modality setting, the paired data $\{(x,y)\}_{i=1}^{n} \sim \mathbb{P}(y|x)$ are given, then, we can directly extract sufficient representation $\phi(x)$ and $\mu(y)$ from spectral decomposition of $\mathbb{P}(y|x)$ in (1) towards to the target tasks. Given the spectral representation, we can sufficient represent $\mathbb{E}[y|x]$, recover $\mathbb{P}(y|x)$, and so on, to complete the particular downstream tasks.

## 3 Spectral Representation

In this section, and the following sections, we will justify the existing methods through the unified spectral framework, which also naturally categorize the SSL methods through their parametriztions, ranging from direct spectral decomposition in Section 3 to energy-based and latent variable spectral representation in Section 4 and Section 5, and introduce the corresponding new algorithms.

### 3.1 Spectral Representation View for Existing SSL

The spectral representation view naturally connects a variety existing SSL algorithms, including contrastive and non-contrastive, which were originally derived from different perspectives with different optimization procedures, but approaching the same target representations.

- **Square Contrastive Representation (HaoChen et al., 2021).** Recall the spectral representation is characterized by (5). However, directly matching the LHS and RHS of (5) may not lead to tractable objective. By multiplying $\tilde{\mathbb{P}}(x)$ to both sides of (5), we obtain

$$\frac{\mathbb{P}(x',x)}{\mathbb{P}(x)\mathbb{P}(x')} = \langle \varphi(x'), \varphi(x) \rangle \Rightarrow \underbrace{\frac{\mathbb{P}(x',x)}{\sqrt{\mathbb{P}(x)\mathbb{P}(x')}}}_{T(x,x')} = \sqrt{\mathbb{P}(x)\mathbb{P}(x')} \langle \varphi(x'), \varphi(x) \rangle. \tag{6}$$

We define the normalized transition operator

$$T(x,x') = \frac{\mathbb{P}(x',x)}{\sqrt{\mathbb{P}(x)\mathbb{P}(x')}}, \tag{7}$$

which plays a key role in further developments below. Concretely, we can now apply mean square matching to $T(x,x')$, leading to the spectral contrastive loss for spectral representation learning, *i.e.*,

$$\min_{\varphi} \ell(\varphi) := \int \left\| T(x,x') - \sqrt{\mathbb{P}(x)\mathbb{P}(x')} \langle \varphi(x'), \varphi(x) \rangle \right\|^2 dx dx' \tag{8}$$

$$= \underbrace{\int T(x,x')^2 dx dx'}_{\text{constant}} - 2\mathbb{E}_{\mathbb{P}(x,x')}\left[\varphi(x)^\top \varphi(x')\right] + \mathbb{E}_{\mathbb{P}(x)\mathbb{P}(x')}\left[\left(\varphi(x)^\top \varphi(x')\right)^2\right], \tag{9}$$

which has been investigated in (HaoChen et al., 2021; Ren et al., 2022b). Obviously, the obtained spectral contrastive loss is compatible with stochastic gradient-based optimizer, therefore, can be solved efficiently.

- **Barlow-Twins Representation (Zbontar et al., 2021).** We consider the linear projected spectral representation, *i.e.*, $\xi(x) = Z^\top \varphi(x)$. Obviously, with any full-rank linear projection $Z$, $\xi(x)$ spans the same subspace as $\varphi(x)$. Particularly, we denote

$$A := \mathbb{E}\left[\varphi(x)\varphi(x)^\top\right] = \int \varphi(x)\varphi(x)^\top \mathbb{P}(x)\,dx = U\Lambda U^\top,$$

where the last equation comes from eigen-decomposition of symmetric $A$ with $U^\top U = I \in \mathbb{R}^{d \times d}$. If we set $Z = U\Lambda^{-1}$, we have the condition to characterize the spectral representation as

$$\int T(x, x') \sqrt{\mathbb{P}(x)} \xi(x) \sqrt{\mathbb{P}(x')} \xi(x')^\top dx dx'$$

$$= \left\langle \int \mathbb{P}(x) \xi(x) \varphi(x) dx, \int \varphi(x') \mathbb{P}(x') \xi(x')^\top dx' \right\rangle = Z^\top A^\top A Z = \mathbf{I}, \tag{10}$$

where the last equation comes from the fact of the definition of $Z$. It implies the learning objective,

$$\min_\xi \left\| \mathbf{I} - \mathbb{E}_{\mathbb{P}(x,x')} \left[ \xi(x) \xi(x')^\top \right] \right\|_2^2. \tag{11}$$

Follow the notation from Zbontar et al. (2021),

$$C = \mathbb{E}_{\mathbb{P}(x,x')} \left[ \xi(x) \xi(x')^\top \right] = \Lambda^{-1} U^\top \mathbb{E}_{\mathbb{P}(x,x')} \left[ \varphi(x) \varphi(x')^\top \right] U \Lambda^{-1} \in \mathbb{R}^{d \times d},$$

where the $\Lambda \in \mathbb{R}^{d \times d}$ is a diagonal matrix and can be estimated through

$$U^\top \mathbb{E} \left[ \varphi(x) \varphi(x)^\top \right] U = \Lambda. \tag{12}$$

Then, we can reformulate (11) equivalently as

$$\min_{U, \varphi} \sum_{i=1}^d (1 - C_{ii})^2 + \sum_{i \neq j} C_{ij}^2, \tag{13}$$

which shares the same solution as

$$\min_{U, \varphi} \sum_{i=1}^d (1 - C_{ii})^2 + \lambda \sum_{i \neq j} C_{ij}^2, \tag{14}$$

recovering the exact Barlow-Twins objective (Zbontar et al., 2021).

**Remark (Optimization Difficulty):** Although we have revealed that the Barlow-Twin formulation is also seeking a spectral representation, the objective in Barlow-Twins (14) is actually not compatible with a stochastic gradient algorithm: the gradient calculation to (14) is biased, and thus, may lead to inferior performance due to the bias induced by approximating the expectation. This observation implies that the Barlow-Twins method also requires a large batch size to reduce the sample bias, even though it is not a contrastive objective, which was the original motivation of Barlow-Twins representation.

- **Variance-Invariance-Covariance Regularization (VICReg) (Bardes et al., 2021).** With the same notations, we consider another linear setup $Z = U\Lambda^{-\frac{1}{2}}$, then, we have $\xi(x) = Z^\top \varphi(x)$ satisfying orthonormal condition, *i.e.*,

$$\mathbb{E} \left[ \xi(x) \xi(x)^\top \right] = Z^\top \mathbb{E} \left[ \varphi(x) \varphi(x)^\top \right] Z = \mathbf{I}. \tag{15}$$

Then, we have the variational characteristic of spectral decomposition w.r.t. $\xi(x)$, *i.e.*,

$$\max_\xi \int T(x, x') \sqrt{\mathbb{P}(x)} \xi(x)^\top \sqrt{\mathbb{P}(x')} \xi(x') dx dx' = \mathbb{E}_{\mathbb{P}(x,x')} \left[ \xi(x)^\top \xi(x') \right]. \tag{16}$$

Combine (15) and (16) with different penalties through the penalty method, we obtain

$$\min_\xi -\mathbb{E}_{\mathbb{P}(x,x')} \left[ \xi(x)^\top \xi(x') \right] + \eta \left( \frac{1}{d} \left( \sum_{i \neq j} \mathrm{Cov}^2(\xi(x)) + \mathrm{Cov}^2(\xi(x')) \right) \right) \tag{17}$$

$$+ \frac{\lambda}{d} \sum_{i=1}^d \max \left( 0, 1 - \sqrt{\mathrm{diag}(\mathrm{Cov}[\xi(x)])} \right) + \frac{\lambda}{d} \sum_{i=1}^d \max \left( 0, 1 - \sqrt{\mathrm{diag}(\mathrm{Cov}[\xi(x')])} \right),$$

which is almost the same as the original VICReg objective proposed in (Bardes et al., 2021), except replacing $-\mathbb{E}_{\mathbb{P}(x,x')}\left[\xi(x)^\top \xi(x')\right]$ with $\mathbb{E}_{\mathbb{P}(x,x')}\left[\|\xi(x) - \xi(x')\|^2\right]$. In fact, recall the fact that under the orthonormal condition (15),

$$\mathbb{E}_{\mathbb{P}(x,x')}\left[\|\xi(x) - \xi(x')\|^2\right] = -2\mathbb{E}_{\mathbb{P}(x,x')}\left[\xi(x)^\top \xi(x')\right] + \underbrace{\mathbb{E}\left[\xi(x)^\top \xi(x)\right]}_{\mathbf{I}} + \underbrace{\mathbb{E}\left[\xi(x')^\top \xi(x')\right]}_{\mathbf{I}},$$

we recovers the VICReg. Similar argument also applies to a variant of VICReg (Balestriero & LeCun, 2022) with square penalty w.r.t. covariance, *i.e.*,

$$\min_\xi \quad -2\mathbb{E}_{\mathbb{P}(x,x')}\left[\xi(x)^\top \xi(x')\right] + \lambda \left\|\mathbb{E}_{\mathbb{P}(x)}\left[\xi(x)\xi(x)^\top\right] - \mathbf{I}\right\|_F^2. \tag{18}$$

**Remark (Optimization Difficulty):** Similar to Barlow-Twins, the optimization objectives of VICReg (17) and (18) also contain terms that are not compatible with stochastic gradient descent algorithm, which may lead to inferior performances due to the bias induced by expectation approximation. This observation implies the VICReg also requires large batch size to reduce the sample bias, even it is not contrastive objective, therefore, inspiring the investigation of alternative penalties to avoid trivial solutions.

- **BYOL (Grill et al., 2020) and MINC Representation (Guo et al., 2025).** BYOL (Grill et al., 2020) and its variants (Tian et al., 2021; Wang et al., 2021; Richemond et al., 2023) can be recast as variants in implementing power iteration for spectral representation as explained in (Guo et al., 2025). We follow the notation and set $Z = U$, then, we have

$$\mathbb{E}\left[\xi(x)\xi(x)^\top\right] = Z^\top \mathbb{E}\left[\varphi(x)\varphi(x)^\top\right] Z = \Lambda, \quad \text{with} \quad \xi(x) = Z^\top \varphi(x) \tag{19}$$

the eigenvalue property that

$$\int T(x, x')\sqrt{\mathbb{P}(x')}\xi(x')^\top dx' \tag{20}$$

$$= \left\langle \sqrt{\mathbb{P}(x)}\varphi(x), \underbrace{\int \varphi(x')\xi(x')^\top \tilde{\mathbb{P}}(x')dx'}_{AZ=U\Lambda} \right\rangle \tag{21}$$

$$= \sqrt{\mathbb{P}(x)}\varphi(x)^\top U\Lambda = \sqrt{\mathbb{P}(x)}\xi(x)^\top \Lambda, \tag{22}$$

which implies

$$\int T(x, x')\sqrt{\mathbb{P}(x')}\xi(x')dx' = \Lambda\sqrt{\mathbb{P}(x)}\xi(x). \tag{23}$$

The fixed point equation (23) of $\xi(x)$ induces the power iteration while ensuring the orthogonality (19) for spectral representation learning by following conceptual rule:

$$\Lambda_{t+1} = \mathbb{E}\left[\xi_t(x)\xi_t(x)^\top\right], \tag{24}$$

$$\Lambda_{t+1}\sqrt{\mathbb{P}(x)}\zeta_{t+1}(x) = \int T(x, x')\sqrt{\mathbb{P}(x')}\xi_t(x')dx', \tag{25}$$

$$\sqrt{\mathbb{P}(x)}\xi_{t+1}(x) = \mathbf{orth}\left(\sqrt{\mathbb{P}(x)}\xi_{t+1}(x)\right), \tag{26}$$

where $\mathbf{orth}(\cdot)$ denotes a conceptual orthogonal operation. The exact integrals in the first two updates and the orthogonalization in the third update are intractable. However, with different variational updates, we arrive at different practical algorithms, including MINC (Guo et al., 2025) and BYOL (Grill et al., 2020).

Concretely, in MINC, exponential moving average (EMA) is adopted for approximating (24), and orthogonal variational version of generalized Hebbian rule (Sanger, 1989; Kim et al., 2005; Xie et al., 2015) for (24)

and (26), which leads to the update

$$\Lambda_{t+1} = \beta\Lambda_t + (1 - \beta)\,\mathbb{E}\left[\xi_t\left(x\right)\xi_t\left(x\right)^\top\right],\tag{27}$$

$$\xi_{t+1}\left(x\right) = \xi_t\left(x\right) + \alpha_t\left(\mathbb{E}_{\mathbb{P}(x,x')}\left[\xi_t\left(x'\right)^\top\nabla_\xi\xi_t\left(x\right)\right] - \mathbb{E}\left[\xi_t\left(x\right)^\top \mathrm{LT}\left[\Lambda_{t+1}\right]\nabla_\xi\xi_t\left(x\right)\right]\right),\tag{28}$$

where the operator $\mathrm{LT}\left(\cdot\right)$ stands for making matrix lower triangular by setting all elements above the diagonal of its matrix argument to zero. As explained in (Guo et al., 2025), the update (28) is obtained by combining Gram-Schmidt process for orthogonalization to the gradient update of

$$\min_\xi\;\int\left[\left\|\int\frac{\mathbb{P}\left(x',x\right)}{\sqrt{\mathbb{P}\left(x\right)}}\xi_t\left(x'\right)dx' - \Lambda_{t+1}\sqrt{\mathbb{P}\left(x\right)}\xi\left(x\right)\right\|_{\Lambda_{t+1}^{-1}}^2\right]dx\tag{29}$$

$$\propto\;-2\mathbb{E}_{\mathbb{P}(x',x)}\left[\xi_t(x')^\top\xi\left(x\right)\right] + \mathbb{E}\left[\xi\left(x\right)^\top\Lambda_{t+1}\xi\left(x\right)\right].\tag{30}$$

Alternatively, BYOL exploits different way to implement the power iteration. Particularly, the $\Lambda$ plays as the eigen-value matrix in stationary solution, therefore, it should satisfy the variational characteristic of eigenvalues as shown in (23). This property induces the learning rule of $\Lambda$ in BYOL. At $t$-th iteration, we have

$$\min_\Lambda\;\int\left[\left\|\int\frac{\mathbb{P}\left(x',x\right)}{\sqrt{\mathbb{P}\left(x\right)}}\xi_t\left(x'\right)dx' - \Lambda\sqrt{\mathbb{P}\left(x\right)}\xi_t\left(x\right)\right\|_2^2\right]dx\tag{31}$$

$$\propto\;-2\mathbb{E}_{\mathbb{P}(x',x)}\left[\xi_t(x')^\top\Lambda\xi_t\left(x\right)\right] + \mathbb{E}\left[\xi_t\left(x\right)^\top\Lambda^\top\Lambda\xi_t\left(x\right)\right]\propto\mathbb{E}_{\mathbb{P}(x',x)}\left[\left\|\Lambda\xi_t\left(x\right) - \xi_t\left(x'\right)\right\|_2^2\right].$$

Meanwhile, the fixed-point condition (23) also induces variational update rule for $\xi$, which is used for BYOL,

$$\min_\xi\;\int\left[\left\|\int\frac{\mathbb{P}\left(x',x\right)}{\sqrt{\mathbb{P}\left(x\right)}}\xi_t\left(x'\right)dx' - \Lambda\sqrt{\mathbb{P}\left(x\right)}\xi\left(x\right)\right\|_2^2\right]dx\tag{32}$$

$$\propto\;-2\mathbb{E}_{\mathbb{P}(x',x)}\left[\xi_t(x')^\top\Lambda\xi\left(x\right)\right] + \mathbb{E}\left[\xi\left(x\right)^\top\Lambda^\top\Lambda\xi\left(x\right)\right]\propto\mathbb{E}_{\mathbb{P}(x',x)}\left[\left\|\Lambda\xi\left(x\right) - \xi_t\left(x'\right)\right\|_2^2\right].$$

Combine (31) and (32), we obtain the original BYOL proposed in Grill et al. (2020).

Similarly, we can also derive other update rule for $\Lambda$ through the properties of eigen-values or Riemannian gradient, as shown in Richemond et al. (2023), leading to a family of BYOL variants. Moreover, Guo et al. (2025) generalizes the technique to derive a series of non-contrastive representation learning, beyond square loss to $f$-mutual information loss.

**Remark (Alternative Fix-Point Iteration):** Besides developing practical fix-point update from (23), we can also exploit the equivalent condition

$$T\left(x,x'\right) = \sqrt{\mathbb{P}\left(x'\right)\mathbb{P}\left(x\right)}\left\langle\xi\left(x'\right),\xi\left(x\right)\right\rangle,\tag{33}$$

which can easily obtained by plugging the definition of $\xi$ into (6). It induces the update via variational implicit method. Specifically, in $t$-th iteration, we have

$$\min_\xi\;\int\left\|T(x,x') - \sqrt{\mathbb{P}\left(x\right)\mathbb{P}\left(x'\right)}\left\langle\xi_t\left(x'\right),\xi\left(x\right)\right\rangle\right\|^2dxdx'\tag{34}$$

$$= \underbrace{\int T(x,x')^2dxdx'}_{\text{constant}} - 2\mathbb{E}_{\mathbb{P}(x,x')}\left[\xi_t\left(x'\right)^\top\xi\left(x\right)\right] + \mathbb{E}_{\mathbb{P}(x)\mathbb{P}(x')}\left[\left(\xi_t(x')^\top\xi\left(x\right)\right)^2\right]\tag{35}$$

$$\propto\;-2\mathbb{E}_{\mathbb{P}(x,x')}\left[\xi_t\left(x'\right)^\top\xi\left(x\right)\right] + \mathbb{E}_{\mathbb{P}(x)}\left[\xi\left(x\right)^\top\underbrace{\mathbb{E}_{\mathbb{P}(x')}\left[\xi_t\left(x'\right)\xi_t\left(x'\right)^\top\right]}_{\Lambda_t}\xi\left(x\right)\right],\tag{36}$$

which induces the same update rule from (29), justifying the correctness of the derivation. Although this condition does not imply new algorithm in direct spectral factorization, we will see that this condition is more convenient for algorithm design in energy-based spectral representation.

**Remark (Efficient Non-Contrastive Learning):** We observe that MINC and BYOL lead to the same spectral representation to square contrastive representation (HaoChen et al., 2021) but by non-contrastive learning. Meanwhile, comparing to the non-contrastive VICReg (Bardes et al., 2021) and Barlow-Twins (Zbontar et al., 2021), MINC and BYOL are compatible with stochastic gradient: one can easily obtain an unbiased stochastic gradient estimator, so they are efficient with any batchsize.

### 3.2 More Spectral Representation Learning

Besides the existing methods for the learning of spectral representation, we here exploit different properties and views of (5) to design several alternatives. We emphasize that the existing methods and the proposed spectral representation learning below are widely applicable to to energy-based, latent-variable, and nonlinear spectral representation, which will be covered in corresponding section later.

- **Unnormalized Distribution Fitting.** Recall the fact that the factorization induces an unnormalized distribution, we can exploit classification-based NCE (Gutmann & Hyvärinen, 2010) or ranking-based NCE (Ma & Collins, 2018) for unnoramlized distribution fitting to implement spectral representation $\varphi(\cdot)$ learning.

  Specifically, for a given $x$, NCE augments the samples from $\mathbb{P}(x'|x)$ with a group of noise samples $\mathbb{P}(x')$, and provides positive labels to samples from $\mathbb{P}(x'|x)$, while negative labels to samples from noise data. Then, we have

  $$\mathbb{P}(y=1|x',x) = \frac{\mathbb{P}(x'|x)}{\mathbb{P}(x'|x) + \mathbb{P}(x')} = \frac{1}{1 + \frac{\mathbb{P}(x')}{\mathbb{P}(x'|x)}}. \tag{37}$$

  Notice that (37) is the solution of binary logistic regression (Gutmann & Hyvärinen, 2012), we can plug the parametrization (5) to (37), and apply logistic regression for spectral representation learning, which leads to

  $$\max_{\varphi} \ \mathbb{E}_{\mathbb{P}(x,x')}\left[\log \frac{\langle \varphi(x'), \varphi(x)\rangle}{1 + \langle \varphi(x'), \varphi(x)\rangle}\right] - \mathbb{E}_{\mathbb{P}(x)\mathbb{P}(x')}\left[\log(1 + \langle \varphi(x'), \varphi(x)\rangle)\right]. \tag{38}$$

  Similarly, the ranking-based NCE (Ma & Collins, 2018), which generalizes binary logistic regression for multi-class classification for unnormalized distribution fitting, can also be used for spectral representation learning. For each $x$, we collect positive sample $x' \sim \mathbb{P}(x'|x)$ and $k-1$ negative sample $x' \sim \mathbb{P}(x')$. Then, the posterior that $k$-th sample is positive denotes as

  $$\mathbb{P}(k|\{x'_i\}_{i=1}^k, x) = \frac{\mathbb{P}(x'_k|x)\prod_{i\neq k}\mathbb{P}(x'_i)}{\sum_j \tilde{\mathbb{P}}(x'_j|x)\prod_{i\neq j}\mathbb{P}(x'_i)} = \frac{\frac{\mathbb{P}(x'_k|x)}{\mathbb{P}(x'_k)}}{\sum_j \frac{\mathbb{P}(x'_j|x)}{\mathbb{P}(x'_j)}}, \tag{39}$$

  which is the solution of multi-class logistic regression. We plug the parametrization (5) to (39) and apply multi-class logistic regression for spectral representation learning, which leads to

  $$\max_{\varphi} \ \mathbb{E}_{\mathbb{P}(x,x')}\left[\log \varphi(x)^\top \varphi(x')\right] - \mathbb{E}_{\mathbb{P}(x)}\left[\log \sum_{x'\sim\mathbb{P}(x')}\left(\varphi(x)^\top \varphi(x')\right)\right]. \tag{40}$$

- **Density Ratio Fitting.** We observe from (6) that $\langle \varphi(x), \varphi(x')\rangle$ play as the density ratio of $\frac{\mathbb{P}(x',x)}{\mathbb{P}(x)\mathbb{P}(x')}$. Therefore, we can exploit the density ratio estimation techniques (Sugiyama et al., 2012; Nguyen et al., 2010; Nachum & Dai, 2020) to learn the spectral representation (Lu et al., 2024).

Specifically, consider the $f$-divergence where $f$ is convex,

$$D_f \left( \mathbb{P}\left(x', x\right) \| \mathbb{P}\left(x\right) \mathbb{P}\left(x'\right) \right) = \mathbb{E}_{\mathbb{P}(x)\mathbb{P}(x')} \left[ f \left( \frac{\mathbb{P}\left(x', x\right)}{\mathbb{P}\left(x\right) \mathbb{P}\left(x'\right)} \right) \right] \tag{41}$$

$$= \sup_g \left( \mathbb{E}_{\mathbb{P}(x', x)} \left[ g\left(x, x'\right) \right] - \mathbb{E}_{\mathbb{P}(x)\mathbb{P}(x')} \left[ f^* \left( g\left(x, x'\right) \right) \right] \right), \tag{42}$$

where $f^*(\cdot)$ denotes the conjugate function of $f(\cdot)$, and the second equation comes from the convexity of $f(\cdot)$ and interchangeability principle (Dai et al., 2017). Meanwhile, by the conjugate property of $f(\cdot)$, the optimal solution to (42) satisfies

$$g\left(x, x'\right) = f' \left( \frac{\mathbb{P}\left(x', x\right)}{\mathbb{P}\left(x\right) \mathbb{P}\left(x'\right)} \right) \Rightarrow \langle \varphi\left(x\right), \varphi\left(x'\right) \rangle = \left(f'\right)^{-1} \left( g\left(x, x'\right) \right), \tag{43}$$

where $f'(\cdot)$ denotes the gradient of $f(\cdot)$. This implies we can consider the parametrization

$$g\left(x, x'\right) = f' \left( \langle \varphi\left(x\right), \varphi\left(x'\right) \rangle \right), \tag{44}$$

and plug into (42) for spectral representation learning (Lu et al., 2024).

For example, in $KL$-divergence, we have $f(u) = u \log u$, $f^*(t) = \exp(t - 1)$, and $f'(u) = \log u + 1$. Therefore, we have

$$g\left(x, x'\right) = \log \left( \langle \varphi\left(x\right), \varphi\left(x'\right) \rangle \right) + 1,$$

with

$$\max_\varphi \; \mathbb{E}_{\mathbb{P}(x', x)} \left[ \log \left( \langle \varphi\left(x\right), \varphi\left(x'\right) \rangle \right) \right] - \mathbb{E}_{\mathbb{P}(x)\mathbb{P}(x')} \left[ \langle \varphi\left(x\right), \varphi\left(x'\right) \rangle \right].$$

In $\chi^2$-divergence, we have $f(u) = (u - 1)^2$, $f^*(t) = \frac{t^2}{4} + t$, and $f'(u) = 2(u - 1)$. Therefore, we have

$$g\left(x, x'\right) = 2 \langle \varphi\left(x\right), \varphi\left(x'\right) \rangle - 2,$$

with

$$\max_\varphi \; 2\mathbb{E}_{\mathbb{P}(x', x)} \left[ \langle \varphi\left(x\right), \varphi\left(x'\right) \rangle \right] - \mathbb{E}_{\mathbb{P}(x)\mathbb{P}(x')} \left[ \left( \langle \varphi\left(x\right), \varphi\left(x'\right) \rangle \right)^2 \right],$$

which is exact the spectral decomposition (9).

- **Generalized Variational Power Iteration.** In BYOL and MINC, we implement the variational fixed-point iteration through (weighted) square loss. We can exploit alternative losses for implementing the update. We consider the condition (3) or (5) for the fixed-point iteration. Concretely, at $t$-th iteration, we are updating $\varphi$ by matching

$$\mathbb{P}\left(x'|x\right) = \mathbb{P}\left(x'\right) \langle \psi_t\left(x'\right), A\psi_{t+1}\left(x\right) \rangle. \tag{45}$$

To obtain a tractable objective, we can exploit either NCE or variational $KL$-divergence we proposed above, which leads to

$$\max_{\psi, A} \; \mathbb{E}_{\mathbb{P}(x, x')} \left[ \log \left( A\psi\left(x\right) \right)^\top \psi_t\left(x'\right) \right] - \mathbb{E}_{\mathbb{P}(x)} \left[ \log \sum_{x' \sim \mathbb{P}(x')} \left( A\psi\left(x\right) \right)^\top \psi_t\left(x'\right) \right], \tag{46}$$

or

$$\max_{\psi, A} \; \mathbb{E}_{\mathbb{P}(x', x)} \left[ \log \left( \left( A\psi\left(x\right) \right)^\top \psi_t\left(x'\right) \right) \right] - \mathbb{E}_{\mathbb{P}(x)\mathbb{P}(x')} \left[ \left( A\psi\left(x\right) \right)^\top \psi_t\left(x'\right) \right]. \tag{47}$$

We emphasize that besides the $KL$-divergence for matching, other variational $f$-divergence in (42) can also be applied for power iteration, which we omit here.

**Remark (Spectral Decomposition and Mutual Information):** If we select $f(u) = u \log u$, we recover the $KL$-divergence between $\mathbb{P}(x, x')$ and $\mathbb{P}(x)\mathbb{P}(x')$, which is known as mutual information, *i.e.*,

$$D_{KL}(\mathbb{P}(x, x') \,||\, \mathbb{P}(x)\mathbb{P}(x')) = I(X, X').$$

There have been several works casting the NCE estimators (38) and (40) as from the Donsker-Varadhan lower bound of mutual information (Hjelm et al., 2018; Tschannen et al., 2019). Here we establish the connection between spectral decomposition and mutual information estimation, which as far as known, has not been revealed formally.

**Remark (Different from SimCLR (Chen et al., 2020)):** We emphasize although SimCLR (Chen et al., 2020) is also derived from ranking-based NCE, it is different from (38) and (40) in the sense that no $\exp(\cdot)$ operation in (38) and (40). We will discuss the $\exp(\cdot)$ effect in energy-based spectral representation in Section 4, which actually induces significant different properties.

**Remark (NCE vs. Square SVD):** Although both (38), (40) and (9) are aiming for spectral representation, the binary NCE (38) and (9) is compatible with stochastic gradient, while the ranking-based NCE (40) requires large-batch to compute second term, due to nonlinearity of log. Therefore, we suggest to use (9) from optimization perspective.

## 4 Energy-based Spectral Representation

In (5), we consider the linear factorization of $\mathbb{P}(x'|x)$. However, one can also consider the energy-based representation (Zhang et al., 2023b; Shribak et al., 2024) for spectral representation in an implicit way.

Specifically, we consider the EBM parametrization

$$\mathbb{P}(x'|x) = \mathbb{P}(x') \exp\left(\upsilon(x')^\top \upsilon(x)\right). \tag{48}$$

we obtain the quadratic potential function, leading to

$$\mathbb{P}(x'|x) = \mathbb{P}(x') \exp\left(\|\upsilon(x')\|^2/2\right) \exp\left(-\|\upsilon(x') - \upsilon(x)\|^2/2\right) \exp\left(\|\upsilon(x)\|^2/2\right). \tag{49}$$

The term $\exp\left(-\frac{\|\upsilon(x') - \upsilon(x)\|^2}{2}\right)$ is the Gaussian kernel, for which we apply the random Fourier feature (Rahimi & Recht, 2007; Dai et al., 2014) and obtain the spectral decomposition of (48),

$$\mathbb{P}(x'|x) = \mathbb{P}(x') \langle \varphi_\omega(x'), \varphi_\omega(x) \rangle_{\mathcal{N}(\omega)}, \tag{50}$$

where $\omega \sim \mathcal{N}(0, \mathbf{I})$, and

$$\varphi_\omega(x) = \exp\left(-\mathbf{i}\omega^\top \upsilon(x)\right) \exp\left(\|\upsilon(x)\|^2/2\right). \tag{51}$$

This bridges the factorized EBMs (48) to infinite-dimensional SVD.

With the energy-based representation (50), we can represent the

$$\mathbb{E}[y|x] = \langle \varphi_\omega(x), W_\omega \rangle_{\mathcal{N}(w)} = \exp\left(\|\upsilon(x)\|^2/2\right) \int \alpha(x') \underbrace{\exp\left(-\|\upsilon(x') - \upsilon(x)\|^2/2\right)}_{k(\upsilon(x'), \upsilon(x))} \tilde{\mathbb{P}}(x')\,dx'.$$

which is linear w.r.t. $\varphi_\omega(x)$, but nonlinear w.r.t. $\upsilon(x)$. In practice, we can either exploit Monte-Carlo approximation for first or second equation to obtain a tractable parametrization for $\mathbb{E}[y|x]$ with parameters $W$ or $\alpha$.

**Remark (Generalized Energy-based Spectral Representation):** The model (48) implictly assumes the partition function is 1, *i.e.*, $\int \mathbb{P}(x') \exp\left(\upsilon(x')^\top \upsilon(x)\right) dx' = 1$, which may not hold in practice, therefore, inducing extra approximation error.

To include the effect of the partition function, we consider

$$\mathbb{P}(x'|x) = \mathbb{P}(x') \exp\left(\upsilon(x')^\top \upsilon(x) - \log Z(x)\right), \quad Z(x) = \int \mathbb{P}(x') \exp\left(\upsilon(x')^\top \upsilon(x)\right) dx'. \quad (52)$$

It is clear that $Z(x)$ can be linearly represented by $\phi_\omega(x)$,

**Proof** We apply the Fourier decomposition to $Z(x)$, *i.e.*,

$$Z(x) = \mathbb{E}_{\mathbb{P}(x')}\left[\exp\left(\upsilon(x')^\top \upsilon(x)\right)\right] = \mathbb{E}_{\mathbb{P}(x')}\left[\langle \varphi_\omega(x), \varphi_\omega(x')\rangle_{\mathcal{N}(\omega)}\right]$$
$$= \langle \phi_\omega(x), \mathbb{E}_{\mathbb{P}(x')}[\varphi_\omega(x')]\rangle_{\mathcal{N}(\omega)} = \langle \varphi_\omega(x), u\rangle_{\mathcal{N}(\omega)}. \quad (53)$$

$\blacksquare$

With this parametrization of $\tilde{\mathbb{P}}(x'|x)$, we have

$$\mathbb{E}[y|x] = \int y\mathbb{P}(y|x)\,dy = \left\langle \frac{\varphi_\omega(x)}{\langle \phi_\omega(x), u\rangle_{\mathcal{N}(\omega)}}, w\right\rangle_{\mathcal{N}(\omega)}. \quad (54)$$

Although the obtained $\varphi_\omega(\cdot)$ is no longer able to represent the $\mathbb{E}[y|x]$ linearly, it is still sufficient for prediction with a nonlinear form. Recall $w$ and $u$ share the same space of $\phi_\omega(x)$, we consider

$$u = \int \mathbb{P}(x')\phi_\omega(x')\,dx' \approx \sum_{i=1}^m \alpha_i\phi_\omega(x'_i) \Rightarrow \langle \phi_\omega(x), u\rangle_{\mathcal{N}(\omega)} \approx \sum_{i=1}^m \alpha_i \exp\left(\upsilon(x'_i)^\top \upsilon(x)\right). \quad (55)$$

Similarly, we have $w \approx \sum_{i=1}^m \beta_i\phi_\omega(x'_i)$. Therefore, we have

$$\mathbb{E}[y|x] \approx \sum_{i=1}^m \frac{\beta_i \exp\left(\upsilon(x'_i)^\top \upsilon(x)\right)}{\sum_{j=1}^m \alpha_j \exp\left(\upsilon(x'_j)^\top \upsilon(x)\right)}, \quad (56)$$

which recovers the attention mechanism, implying we can exploit attention for downstream tasks.

On the other hand, we can represent $\mathbb{P}(y|x) \propto \exp\left(\upsilon(x)^\top \eta(y)\right)$ by energy-based model.

**Proof** We consider the Fourier decomposition of $\mathbb{P}(z|x)$,

$$\mathbb{P}(z|x) = \frac{1}{Z(x)} \exp\left(\upsilon(x)^\top W_z + b_z\right) = \frac{1}{Z(x)} \langle \varphi_\omega(x), \psi_\omega(z)\rangle_{\mathcal{N}(\omega)}. \quad (57)$$

Therefore, we have

$$\mathbb{P}(y|x) = \int \mathbb{P}(y|z)\mathbb{P}(z|x)\,dz = \frac{1}{Z(x)}\left\langle \varphi_\omega(x), \underbrace{\int \mathbb{P}(y|z)\psi_\omega(z)\,dz}_{\zeta_\omega(y)}\right\rangle_{\mathcal{N}(\omega)} \propto \exp\left(\nu(x)^\top \eta(y)\right). \quad (58)$$

$\blacksquare$

### 4.1 Energy-based Spectral Representation Learning

With this connection between energy-based parametrization and spectral representation, we can unify a variety of existing SSL algorithms into spectral representation framework, which originally developed from different perspective.

- **Simple Framework for Contrastive Learning of Representation (SimCLR) (Chen et al., 2020).** With the structure of $\mathbb{P}(x'|x)$ in (52), one can apply the ranking-based noise contrastive estimation (NCE) (Gutmann & Hyvärinen, 2010; Ma & Collins, 2018) for $\nu(x)$, which induces the spectral representation $\varphi(x)$ through (51).

  Specifically, following the notation in Section 3.2, we plug the energy-based parametrization (48) or the normalized version (52) into the posterior formulation (39), which leads to

$$p_\theta\left(k|\left\{x'\right\}_{i=1}^k, x\right) = \frac{\exp\left(\upsilon\left(x'_k\right)^\top \upsilon(x)\right)}{\sum_{j=1} \exp\left(\upsilon\left(x'_j\right)^\top \upsilon(x)\right)}, \tag{59}$$

  We apply multi-class logistic regression upon (59), *i.e.*, minimizing the *KL*-divergence between $\mathbb{P}\left(k|\left\{x'\right\}_{i=1}^k, x\right)$ and $p_\theta\left(k|\left\{x'\right\}_{i=1}^k, x\right)$, leading to

$$\min_\upsilon \ -\mathbb{E}_{\mathbb{P}(x,x')}\left[\upsilon\left(x'\right)^\top \upsilon(x)\right] + \mathbb{E}_{\mathbb{P}(x)}\left[\log\sum_{i=1}\exp\left(\nu\left(x'_i\right)^\top \upsilon(x)\right)\right]. \tag{60}$$

  If we restrict $\|\upsilon(x)\|_2 = 1$ by considering the parametrization $\frac{\upsilon(x)}{\|\upsilon(x)\|_2}$, we obtain exact the SimCLR (Chen et al., 2020).

  Besides revealing the relationship between SimCLR and spectral representation, more importantly, we identify the correct parametrization of regressor or classifier for downstream tasks in terms of energy-based spectral representation $\upsilon(x)$, as we discussed in (56).

- **Word2Vec (Mikolov et al., 2013; Levy & Goldberg, 2014; Goldberg & Levy, 2014).** When we apply binary NCE with nonparametric model to text, we obtain word2vec. Concretely, we have the parametrization for word-context pair $(w, c)$ as

$$\frac{\mathbb{P}(w, c)}{\mathbb{P}(w)\mathbb{P}(c)} = \exp\left(\phi(w)^\top \phi(c)\right). \tag{61}$$

  Apply binary NCE to (61) with linear parametrization, *i.e.*, $\phi(w) = Vw$ and $\phi(c) = Vc$ for observed $(w, c) \sim \mathcal{D}$, we obtain word2vec,

$$\max_W \ \mathbb{E}_{\mathbb{P}(w,c)}\left[\log\sigma\left(w^\top V^\top Vc\right)\right] - \mathbb{E}_{\mathbb{P}(w)\mathbb{P}(c)}\left[\log\sigma\left[-w^\top V^\top Vc\right]\right], \tag{62}$$

  where $\sigma(\cdot) = \frac{1}{1+\exp(\cdot)}$.

- **Momentum Contrast Representation Learning (MoCo) (He et al., 2020).** We can also recover MoCo (He et al., 2020) as a variational form of power iteration for energy-based spectral representation. As we discussed in Section 3.1, the power iteration can be equivalently derived based on the fixed-point iteration of (33). We exploit the idea in energy-based spectral representation. Concretely, we start with the spectral decomposition (50) of the energy-based model of (52), *i.e.*,

$$\mathbb{P}(x'|x) = \frac{1}{Z(x)}\mathbb{P}(x')\left\langle\varphi_\omega(x'), \varphi_\omega(x)\right\rangle_{\mathcal{N}(\omega)}. \tag{63}$$

  We emphasize although we derive the algorithm based on (52), the derivation also applicable to (48), which leads to the same updates.

Following the fixed-point iteration idea, at $t$-th iteration, we propose to match $\mathbb{P}(x'|x)$ and $\frac{1}{Z(x)}\mathbb{P}(x')\langle\varphi_{\omega,t}(x'),\varphi_\omega(x)\rangle_{\mathcal{N}(\omega)}$, with $\varphi_{\omega,t}(x')$ fixed as previous iteration solution and one $\varphi_\omega(x)$ updated. To obtain a tractable update, recall the fact that the RHS of (63) is EBM, we exploit ranking-based NCE for model learning to avoid the partition function calculation, instead of $L_2$ distance in (34), leading to

$$\max_\varphi \ \mathbb{E}_{\mathbb{P}(x,x')}\left[\log\langle\varphi_{\omega,t}(x'),\varphi_\omega(x)\rangle_{\mathcal{N}(\omega)}\right] - \mathbb{E}_{\mathbb{P}(x)}\left[\log\sum_{x'\sim\mathbb{P}(x')}\left(\langle\varphi_{\omega,t}(x'),\varphi_\omega(x)\rangle_{\mathcal{N}(\omega)}\right)\right]. \quad (64)$$

Based on the random feature property, *i.e.*,

$$\langle\varphi_{\omega,t}(x'),\varphi_\omega(x)\rangle_{\mathcal{N}(\omega)} = \exp\left(\upsilon_t(x')^\top \upsilon(x)\right),$$

we achieve the equivalent objective for $\upsilon$,

$$\upsilon_{t+1}(x) = \operatorname*{argmin}_\upsilon \ -\sum_{x\sim\mathcal{D}}\left[\upsilon(x'_k)^\top \upsilon_t(x) - \log\sum_{i=1}\exp\left(\upsilon(x'_i)^\top \upsilon_t(x)\right)\right], \quad (65)$$

which recover the MoCo learning objective (He et al., 2020).

- **Diffusion Spectral Representation (Shribak et al., 2024).** We revisit the EBM understanding of the diffusion model, which paves the path for efficient learning of spectral representation (52) through diffusion view, establishing the connection to (Shribak et al., 2024).

Given $x$, we consider the samples from $x'\sim\mathbb{P}(x'|x)$ is perturbed with Gaussian noise, *i.e.*, $\mathbb{P}(\tilde{x}'|x';\beta) = \mathcal{N}\left(\sqrt{1-\beta}x',\beta\sigma^2 I\right)$. Then, we parametrize the corrupted dynamics as

$$\mathbb{P}(\tilde{x}'|x;\beta) = \int \mathbb{P}(\tilde{x}'|x';\beta)\mathbb{P}(x'|x)dx' = \int \mathbb{P}(\tilde{x}'|x';\beta)\mathbb{P}(x')\exp\left(\upsilon(x)^\top \upsilon(x')\right)dx' \quad (66)$$

$$= \left\langle \varphi_\omega(x), \underbrace{\int \mathbb{P}(\tilde{x}'|x';\beta)\mathbb{P}(x')\varphi_\omega(x')\,dx'}_{\xi_\omega(\tilde{x}',\beta)} \right\rangle_{\mathcal{N}(\omega)} \propto \exp\left(\upsilon(x)^\top \mu(\tilde{x}',\beta)\right), \quad (67)$$

in which third equation comes from Fourier decomposition, and the four equation comes from inverse Fourier transformation. We denote $\mu(\tilde{x},\beta)$ as the reparametrization.

**Proposition 1 (Tweedie's Identity (Efron, 2011))** *For arbitrary corruption* $\mathbb{P}(\tilde{x}'|x';\beta)$ *and* $\beta$ *in* $\mathbb{P}(\tilde{x}'|x;\beta)$*, we have*

$$\nabla_{\tilde{x}'}\log\mathbb{P}(\tilde{x}'|x;\beta) = \mathbb{E}_{\mathbb{P}(x'|\tilde{x}',x;\beta)}\left[\nabla_{\tilde{x}'}\log\mathbb{P}(\tilde{x}'|x';\beta)\right]. \quad (68)$$

This can be easily verified by simple calculation, *i.e.*,

$$\nabla_{\tilde{x}'}\log\mathbb{P}(\tilde{x}'|x;\beta) = \frac{\nabla_{\tilde{x}'}\mathbb{P}(\tilde{x}'|x;\beta)}{\mathbb{P}(\tilde{x}'|x;\beta)} = \frac{\nabla_{\tilde{x}'}\int \mathbb{P}(\tilde{x}'|x';\beta)\mathbb{P}(x'|x)dx'}{\mathbb{P}(\tilde{x}'|x;\beta)}$$

$$= \int \frac{\nabla_{\tilde{x}'}\log\mathbb{P}(\tilde{x}'|x';\beta)\mathbb{P}(\tilde{x}'|x';\beta)\mathbb{P}(x'|x)}{\mathbb{P}(\tilde{x}'|x;\beta)}dx' = \mathbb{E}_{\mathbb{P}(x'|\tilde{x}',x;\beta)}\left[\nabla_{\tilde{x}'}\log\mathbb{P}(\tilde{x}'|x';\beta)\right]. \quad (69)$$

For Gaussian perturbation with (66), Tweedie's identity (68) is applied as

$$\upsilon(x)^\top \nabla_{\tilde{x}'}\mu(\tilde{x}',\beta) = \mathbb{E}_{\mathbb{P}(x'|\tilde{x}',x;\beta)}\left[\frac{\sqrt{1-\beta}x'-\tilde{x}'}{\beta\sigma^2}\right]$$

$$\Rightarrow \tilde{x}' + \beta\sigma^2\upsilon(x)^\top \nabla_{\tilde{x}'}\mu(\tilde{x}',\beta) = \sqrt{1-\beta}\mathbb{E}_{\mathbb{P}(x'|\tilde{x}',s,a;\beta)}[x']. \quad (70)$$

Let us introduce new reparametrization $\zeta(\tilde{x}';\beta) = \nabla_{\tilde{x}'}\mu(\tilde{x}',\beta)$, since we only need $\nu(x)$ for energy-based spectral representation, we can learn $\upsilon(x)$ and $\zeta(\tilde{x}';\beta)$ by matching both sides of (70),

$$\min_{\upsilon,\zeta} \ \mathbb{E}_\beta\mathbb{E}_{(x,\tilde{x}')}\left[\left\|\tilde{x}' + \beta\sigma^2\upsilon(x)^\top \zeta(\tilde{x}',\beta) - \sqrt{1-\beta}\mathbb{E}_{\mathbb{P}(x'|\tilde{x}',x;\beta)}[x']\right\|^2\right] \quad (71)$$

which shares the same optimum of

$$\min_{\nu,\zeta} \ \ell_{\text{diff}}(\nu,\zeta) := \mathbb{E}_\beta \mathbb{E}_{(x,\tilde{x}',x')} \left[ \left\| \tilde{x}' + \beta\sigma^2 \upsilon(x)^\top \zeta(\tilde{x}',\beta) - \sqrt{1-\beta}x' \right\|^2 \right]. \tag{72}$$

Obviously, we recovered the diffusion loss for spectral representation learning (Shribak et al., 2024).

## 4.2 More Energy-based Spectral Representation Learning

Besides the existing algorithms for energy-based spectral representation we explained above, we consider the instantiations of the algorithms proposed Section 3.2 for energy-based spectral representation. As we revealed, SimCLR is the instantiation of NCE (40) applied for energy-based parametrization (48) and (52), and MoCo is the instantiation of power iteration (65) applied for energy-based parametrization. Here we introduce the instantiation of density ratio estimation for energy-based spectral representation. However, we emphasize that there are more variants with different divergences can be instantiated by plugging energy-based representation into corresponding objectives proposed in Section 3.2.

- **Density Ratio Fitting.** We plug the energy-based representation (48) into (42) with $KL$-divergence, leading to

$$\max_\upsilon \ \mathbb{E}_{\mathbb{P}(x',x)}\left[\langle \upsilon(x), \upsilon(x')\rangle\right] - \mathbb{E}_{\mathbb{P}(x)\mathbb{P}(x')}\left[\exp\left(\langle \upsilon(x), \upsilon(x')\rangle\right)\right], \tag{73}$$

  which shares some similarity to ranking-based NCE (40), but with the major difference lies in the negative term is expectation of exp, rather than exp of expectation, making the optimization much easier.

  We also emphasize that the density ratio estimation is only applicable for (48). If the EBM with normalization factor introduced (52), the density ratio fitting will lead to intractable optimization with the explicit calculation of $Z(x)$.

# 5 Latent-Variable Spectral Representation

Besides the linear factorization and energy-based parametrization for spectral representation in Section 3 and Section 4, respectively, in this section, we reveal the latent-variable model as an alternative implicit spectral representation parametrization.

Recall the justification of the sufficiency of spectral representation for representation $\mathbb{E}[y|x]$ for any $\mathbb{P}(y|x)$, *i.e.*,

$$\mathbb{E}[y|x] = \int y\mathbb{P}(y|x)\,dy = \int \underbrace{\left(\int y\mathbb{P}(y|z)\,dy\right)}_{w(z)} \mathbb{P}(z|x)\,dz = \langle w(\cdot), \mathbb{P}(\cdot|x)\rangle_{L_2}, \tag{74}$$

where the second equation comes from law of total probability. It implies that $\mathbb{P}(z|x)$ can also be the sufficient representation. In fact, $\mathbb{P}(z|x)$ also provides us a non-negative spectral decomposition of $\mathbb{P}(x'|z)$, *i.e.*,

$$\mathbb{P}(x'|x) = \int \mathbb{P}(x'|z)\,\mathbb{P}(z|x)\,dz = \int \frac{\mathbb{P}(z|x')\,\mathbb{P}(x')}{\mathbb{P}(z)}\mathbb{P}(z|x)\,dz \tag{75}$$

$$= \mathbb{P}(x')\left\langle \sqrt{\Lambda(z)}\mathbb{P}(z|x'), \sqrt{\Lambda(z)}\mathbb{P}(z|x)\right\rangle_{L_2}, \tag{76}$$

where $\Lambda(z) = \frac{1}{\mathbb{P}(z)}$ is a diagonal operator, therefore, $\sqrt{\Lambda(z)}\mathbb{P}(z|x)$ spans the same function space of $\mathbb{P}(z|x)$.

**Remark (Tractable Predictor):** We demonstrate the $\mathbb{P}(z|x)$ as the sufficient representation for linearly representing $\mathbb{E}[y|x]$ in (74). The representation in (74) requires an integration. For discrete latent variable $z \in \{1,\ldots,k\}$, this can be exactly calculated as

$$\mathbb{E}[y|x] = \langle w(\cdot), \mathbb{P}(\cdot|x)\rangle_{L_2} = \sum_{i=1}^{k} w_i \mathbb{P}(z_i|x).$$

However, for general continuous latent variable distribution, the integration might not be tractable in general cases. Therefore, we may need to consider the approximator. One idea is following Monte-Carlo approximation. With samples $\{z_i\}_{i=1}^n \sim \mathbb{P}(z|x)$, we have

$$\mathbb{E}[y|x] = \langle w(\cdot), \mathbb{P}(\cdot|x) \rangle_{L_2} \approx \frac{1}{n} \sum_{i=1}^n w(z_i). \tag{77}$$

Particularly, if we have softmax parametrization for $\mathbb{P}(z|x)$, *i.e.*,

$$\mathbb{P}(z|x) = \frac{\exp\left(v(x)^\top W_z + b_z\right)}{\sum_{z=1}^k \exp\left(v(x)^\top W_z + b_z\right)}, \tag{78}$$

which is an EBM. Based on our previous argument in Section 4, we can also parametrize the $\mathbb{P}(y|x) \propto \exp\left(v(x)^\top \eta(y)\right)$ through energy-based model. Particularly, if it is classification problem, we have

$$\mathbb{P}(y|x) = \frac{\exp\left(v(x)^\top W_y + b_y\right)}{\sum_{z=1}^k \exp\left(v(x)^\top W_y + b_y\right)}.$$

This implies for classification problem, we can consider softmax for $\mathbb{P}(y|x)$; while for regression problem, we can consider the Monte-Carlo approximation from EBM (56).

**Remark (Expressiveness):** In fact, the latent variable representation $\mathbb{P}(z|x)$ is a special case of linear sufficient spectral representation with additional distribution constraint, *i.e.*, $\tilde{\mathbb{P}}(z|x) \geqslant 0$ and $\int \tilde{\mathbb{P}}(z|x)\,dz = 1$. This connection can be revealed more clearly if the latent variable is discrete. Concretely, if $z \in \{1, \ldots, d\}$, then, $\mathbb{P}(\cdot|x) \in \mathbb{R}^d$ is a $d$-dimensional vector in simplex. Such constrained simplex representation limits the model expressiveness, as demonstrated in (Agarwal et al., 2020, Proposition 2): there exists $\mathbb{P}(x'|x)$ has finite rank $d^2$, but requires at least $2^{\Omega(d)}$ finite latent variable $z$ (Yannakakis, 1988; Rothvoß, 2017), showing the original spectral representation can be more compact with fewer dimensions.

However, on the other hand, the density constraint also brings the benefits in terms of modeling and learning that we can exploit latent variable model for representation, which can be infinite-dimensional with continuous latent variable $z$ and be fitted through probabilistic model learning, which we will discuss below.

### 5.1 Latent-Variable Spectral Representation Learning

By revealing the spectral representation interpretation of latent-variable model, we can recast a variety of existing representation learning into our spectral representation framework.

- **DeepCluster (Caron et al., 2018).** We here discuss the connection of DeepCluster (Caron et al., 2018) as latent-variable spectral representation learning. Specifically, we consider the softmax parametrization of $\mathbb{P}(z|x)$ in (78). Then, it can be equivalently written as

$$\mathbb{P}(z|x) \propto \exp\left(-\frac{1}{2\sigma^2} \|Cz - v(x)\|_2^2\right), \tag{79}$$

with $W_z = \frac{Cz}{\sigma^2}$, $b_z = \frac{1}{\sigma^2} z^\top C^\top C z$, $z$ as one-hot vector, *i.e.*, $z \in \{0,1\}^k$, $z^\top \mathbf{1} = 1$, and $C \in \mathbb{R}^{d \times k}$.

Given samples $x \sim \mathcal{D}$, it is natural to consider the maximum likelihood to the latent variable model, *i.e.*,

$$\max_{\mathbb{P}(z|x)} \mathbb{E}_{\mathcal{D}}\left[\log \int \mathbb{P}(x,z)\,dz\right] = \max_{\mathbb{P}(z|x)} \max_{q(z|x)} \mathbb{E}_{\mathcal{D}}\left[\mathbb{E}_{q(z|x)}\left[\log \mathbb{P}(x,z)\right] + H(q(z|x))\right] \tag{80}$$

$$= \max_{\mathbb{P}(z|x)} \max_{q(z|x)} \mathbb{E}_{\mathcal{D}}\left[\mathbb{E}_{q(z|x)}\left[\log \mathbb{P}(z|x)\,\mathbb{P}(x)\right] + H(q(z|x))\right], \tag{81}$$

where the first equation comes from the Jensen's inequality, and the second equation comes from definition of joint probability.

Applying EM algorithm for optimizing (80), we recover the DeepCluster (Caron et al., 2018), *i.e.*,

  – **E-step.** The nonparametrized latent variable distribution $q(z_i|x_i)$ for each data $x_i$ is obtained by optimizing (80) with fixed $\mathbb{P}(z|x)$ and nonparametric $\mathbb{P}(x)$. In DeepCluster (Caron et al., 2018), this is approximated $k$-means, which can be understood as pushing zero variance in the mixture of Gaussians in $q(z|x)$ (Bishop & Nasrabadi, 2006), leading to

$$\min_C \frac{1}{n} \sum_{i=1}^{n} \min_{z_i \in \{0,1\}^k, z_i^\top \mathbf{1} = 1} \|C z_i - \upsilon(x_i)\|_2^2. \tag{82}$$

  – **M-step.** Plug the learned $\{z_i\}_{i=1}^n$ and the parametrization of $\mathbb{P}(z|x)$ in (79) and nonparametric $\mathbb{P}(x)$ into (81) for representation learning, we obtain

$$\max_{\upsilon} \widehat{\mathbb{E}}_{x_i \sim \mathcal{D}} \left[ \upsilon(x_i)^\top W_{z_i} + b_{z_i} - \log \sum_{z=1}^{k} \exp\left( \upsilon(x_i)^\top W_{z_i} + b_{z_i} \right) \right]. \tag{83}$$

We emphasize that although we reveal the connection between $C$ and $(W, b)$ in (79), such relationship has not been exploited in DeepCluster, and the $C$ and $W$ are two independent variables.

- **Self-Labeling (SeLa) (Asano et al., 2019).** The Self-Labelling (SeLa) (Asano et al., 2019) also seeks latent-variable spectral representation, but with different parametrization for $q(z|x)$, and thus different learning updates. Specifically, with the nonparametrc parametrization of $q(z|x) \in \Delta(Z)$, $Z \in \mathbb{R}^k$, where $z \in \{1, \ldots, k\}$ is finite-dimensional latent variable, for each sample $x \sim \mathcal{D}$, *i.e.*, $q(\cdot|x)$ is a $k$-dimensional non-negative vector with constraint $q(\cdot|x)^\top \mathbf{1} = 1$, we have

$$q(z, x) = q(z|x)\mathbb{P}(x) = q(z|x)\frac{1}{|\mathcal{D}|}.$$

Meanwhile, we consider uniform joint distribution as prior $p(z, x) = \frac{1}{k \cdot |\mathcal{D}|}$, we have

$$KL(q(z, x) \| p(z, x)) = \widehat{\mathbb{E}}_{x_i \sim \mathcal{D}} \mathbb{E}_{q(z|x_i)} \left[ \log q(z|x) - \log \frac{1}{k} \right].$$

With these parametrization, the variational maximum likelihood of latent variable model (81) can be equivalently written as

$$\max_{\upsilon} \max_{q(z,x)} \langle q(z, x), \log \mathbb{P}(z|x) \rangle - KL(q(z, x) \| p(z, c)). \tag{84}$$

Obviously, (84) is exactly the objective of SeLa (Asano et al., 2019) if we exploit softmax posterior $\mathbb{P}(z|x)$ in (78).

Particularly, to avoid the potential numerical degeneracy, in SeLa, the posterior $q(z|x)$ is further constrained as

$$\sum_{x \in \mathcal{D}} q(z, x) = \frac{1}{k}, \forall z \in \{1, \ldots, k\}. \tag{85}$$

Therefore, we conclude that the original SeLa is optimizing (84) with softmax parametrization of $\mathbb{P}(z|x)$ and nonparametric $q(z|x)$ with additional constraints. For the optimization, SeLa follows the same **M-step** as DeepCluster (Caron et al., 2018), but different **E-step** for $q(z, x)$ in (84) as an optimal transport problem.

- **DINO (Caron et al., 2021) and SwAV (Caron et al., 2020).** The DIstillation with NO label (DINO) (Caron et al., 2021; Oquab et al., 2023; Siméoni et al., 2025) and Swapping Assignments

between multiple Views (SwAV) (Caron et al., 2020) can both be recast as latent-variable spectral representation. Recall the spectral property of $\mathbb{P}(z|x)$,

$$\int \mathbb{P}(z|x')\,\mathbb{P}(x'|x)\,dx' = \mathbb{P}(z|x),\tag{86}$$

which can be verified by simple calculation, *i.e.*,

$$\begin{aligned}\int \mathbb{P}(z|x')\,\mathbb{P}(x'|x)\,dx' &= \int \mathbb{P}(z|x')\int \mathbb{P}(x'|y)\,\mathbb{P}(y|x)\,dydx'\\&= \int \left(\int \mathbb{P}(z|x')\,\mathbb{P}(x'|y)\,dx'\right)\mathbb{P}(y|x)\,dy\\&= \int \mathbb{P}(z|y)\,\mathbb{P}(y|x)\,dy = \mathbb{P}(z|x).\end{aligned}$$

Then, we can learn the $\mathbb{P}(z|x)$, so the representation, by matching the LHS and RHS of (86) via a variety of divergences over distributions, *e.g.*, $f$-divergence,

$$\min_{\mathbb{P}(z|x)}\ \mathbb{E}_{x,x'}\mathbb{E}_{\mathbb{P}(z|x')}\left[f\left(\frac{\mathbb{P}(z|x)}{\int \mathbb{P}(z|x')\,\mathbb{P}(x'|x)\,dx'}\right)\right].\tag{87}$$

However, the learning objective (87) is difficult to be optimized, due to the intractable integration $\int \mathbb{P}(z|x')\,\mathbb{P}(x'|x)\,dx'$ inside of a nonliner $f(\cdot)$. One can exploit power iteration trick, where we fix $\mathbb{P}(z|x')$ in LHS of (86), and update RHS through distribution matching (87). Concretely, at $t$-th iteration, we update $\mathbb{P}z|x$ through

$$\int \mathbb{P}_t(z|x')\,\mathbb{P}(x'|x)\,dx' = \mathbb{P}_{t+1}(z|x).\tag{88}$$

We take the $KL$-divergence as an example with $f(u) = u\log u$ for matching (88). Then, the $t$-th iteration update for $\mathbb{P}_{t+1}(z|x)$ can be written as a variational optimization, *i.e.*,

$$\min_{\mathbb{P}(z|x)}\ \mathbb{E}_{x,x'}\mathbb{E}_{\mathbb{P}_t(z|x')}\left[\log \mathbb{P}(z|x)\right].\tag{89}$$

Similarly, we can also exploit

$$\min_{\mathbb{P}(z|x')}\ \mathbb{E}_{x,x'}\mathbb{E}_{\mathbb{P}_t(z|x)}\left[\log \mathbb{P}(z|x')\right].\tag{90}$$

Plug the softmax parametrized multinomial distribution of $\mathbb{P}(z|x)$ in (78) to the power iteration matching of stationary latent variable model (89) and (90), we obtain

$$\max_v \widehat{\mathbb{E}}_{x,x'\sim\mathcal{D}}\mathbb{E}_{\mathbb{P}_t(z|x')}\left[v(x)^\top Wz + b_z - \log\sum_{z=1}^k \exp\left(v(x)^\top Wz + b_z\right)\right]$$

$$+ \widehat{\mathbb{E}}_{x,x'\sim\mathcal{D}}\mathbb{E}_{\mathbb{P}_t(z|x)}\left[v(x')^\top Wz + b_z - \log\sum_{z=1}^k \exp\left(v(x')^\top Wz + b_z\right)\right].\tag{91}$$

recovering the exact objective of DINO (Caron et al., 2021) and SwAV (Caron et al., 2020).

The major difference between DINO and SwAV lies in the implementation of $\mathbb{P}_t(z|x)$ in (91): In DINO, $\mathbb{P}_t(z|x)$ is simply exploiting the model in previous iteration; while in SwAV exploits the optimal transport solution to (84), the same as **E-step** in SeLa (Asano et al., 2019).

## 5.2    More Latent-Variable Spectral Representation Learning

We can also exploit the algorithms in Section 3.2 for latent-variable spectral representation, which we omit here. We mainly focus on exploiting the distribution property of latent-variable model for more learning algorithms.

- **Variational ELBO for Multi-View.** Recall in (76) that we have

$$\mathbb{P}\left(x'|x\right) = \mathbb{P}\left(x'\right) \int \frac{\mathbb{P}\left(z|x'\right)\mathbb{P}\left(z|x\right)}{\mathbb{P}\left(z\right)} dz.$$

Given samples $(x, x') \sim \mathcal{D}$, we consider the maximum likelihood, *i.e.*,

$$\max_{\mathbb{P}(z|x'),\mathbb{P}(z|x)} \mathbb{E}_{x,x'\sim\mathcal{D}} \left[\log \int \frac{\mathbb{P}\left(z|x'\right)\mathbb{P}\left(z|x\right)}{\mathbb{P}\left(z\right)} dz + \log \mathbb{P}\left(x'\right)\right] \tag{92}$$

$$\propto \max_{\mathbb{P}(z|x'),\mathbb{P}(z|x),\mathbb{P}(z)} \max_{q(z|x,x')} \mathbb{E}_{x,x'\sim\mathcal{D}} \mathbb{E}_{q(z|x,x')} \left[\log \frac{\mathbb{P}\left(z|x'\right)\mathbb{P}\left(z|x\right)}{\mathbb{P}\left(z\right)} - \log q\left(z|x,x'\right)\right]. \tag{93}$$

With different parametrization and learning of $q\left(z|x,x'\right)$, we can obtain different latent-variable spectral representation learning by optimizing (92). For example, if we exploit nonparamtric $q\left(z|x,x'\right)$, we obtain an optimal transport optimization for cluster assignment. With the cluster fixed, we can update $\mathbb{P}\left(z|x'\right), \mathbb{P}\left(z|x\right), \mathbb{P}\left(z\right)$ by stochastic gradient descent.

- **Contrastive Divergence for Restricted Boltzmann Machine.** With softmax parametrization of $\mathbb{P}\left(z|x\right)$ is actually the Gaussian-Softmax restricted Boltzmann machine,

$$\mathbb{P}\left(z|x\right) = \frac{\exp\left(\upsilon(x)^\top W_z + b_z\right)}{\sum_{z=1}^k \exp\left(\upsilon(x)^\top W_z + b_z\right)}, \tag{94}$$

$$\mathbb{P}\left(\upsilon(x)|z\right) = \mathcal{N}\left(Cz, \sigma^2 \mathbf{I}\right), \tag{95}$$

where $W_z = \frac{Cz}{\sigma^2}$, $b_z = \frac{1}{\sigma^2}z^\top C^\top Cz$, $z$ as one-hot vector, *i.e.*, $z \in \{0,1\}^k$, $z^\top \mathbf{1} = 1$, and $C \in \mathbb{R}^{d\times k}$. The parametrization also induces the distributions

$$\mathbb{P}\left(x,z\right) = \frac{\exp\left(\upsilon(x)^\top W_z + b_z\right)}{\int \sum_z \exp\left(\upsilon\left(x\right)^\top Wz\right) dx}, \tag{96}$$

with the marginal distribution

$$\mathbb{P}\left(x\right) = \frac{\sum_z \exp\left(\upsilon(x)^\top W_z + b_z\right)}{\int \sum_z \exp\left(\upsilon\left(x\right)^\top Wz\right) dx}, \tag{97}$$

which is essentially a mixture of Gaussians w.r.t. $\upsilon\left(x\right)$. Then, we have the MLE as

$$\max_{W,\upsilon} \ell\left(W,\upsilon\right) = \mathbb{E}_x \left[\log \sum \exp\left(\upsilon\left(x\right)^\top Wz\right) - \log \int \sum_z \exp\left(\upsilon\left(x\right)^\top Wz\right) dx\right], \tag{98}$$

with the gradients

$$\nabla_\upsilon \ell\left(W,\upsilon\right) = \mathbb{E}_x \mathbb{E}_{\mathbb{P}(z|x)} \left[\nabla \upsilon(x)^\top Wz\right] - \mathbb{E}_{\mathbb{P}(z,x)} \left[\nabla \upsilon(x)^\top Wz\right], \tag{99}$$

$$\nabla_W \ell\left(W,\upsilon\right) = \mathbb{E}_x \mathbb{E}_{\mathbb{P}(z|x)} \left[\nabla_W \upsilon(x)^\top Wz\right] - \mathbb{E}_{\mathbb{P}(z,x)} \left[\nabla_W \upsilon(x)^\top Wz\right]. \tag{100}$$

Due to the restricted Boltzmann machine structure, we can sample $(x,z) \sim \mathbb{P}\left(z,x\right)$ by blockwise Gibbs sampling, *i.e.*,

  - **Sample $z$.** $\tilde{\mathbb{P}}\left(z|x\right)$ is a multinomial distribution, therefore, sampling $z \sim \tilde{\mathbb{P}}\left(z|x\right)$ is relative simple.
  - **Sample $x$.** Sampling $x$ is relatively difficult as we have $\tilde{\mathbb{P}}\left(x|z\right) = \frac{\exp\left(\upsilon(x)^\top Wz\right)}{\int \exp\left(\upsilon(x)^\top Wz\right)dx}$ as a general EBM, therefore, we can exploit the Langevin sampling strategy, *i.e.*,

$$x_{t+1} = x_t + \alpha_t \nabla_x \upsilon\left(x\right)^\top Wz + \sqrt{2\eta_t}\epsilon, \quad \text{with} \quad \epsilon \sim \mathcal{N}\left(0, I\right).$$

Alternatively, we propose another sampling strategy by exploiting the fact that

$$\tilde{\mathbb{P}}\left(\upsilon(x)|z\right) = \mathcal{N}\left(Cz, \sigma^2 I\right).$$

Specifically, we first sample $y \sim \mathcal{N}\left(Cz, \sigma^2 I\right)$. Then, we extract

$$x = \operatorname*{argmin}_x \|y - \upsilon\left(x\right)\|^2.$$

With the sampling strategy, we can approximate $\ell\left(W, \upsilon\right)$ in (99), which is an instantiation of contrastive divergence (CD) algorithm (Hinton, 2002) for representation learning.

- **Adversarial Training.** We acknowledge the difficulty in directly optimizing (87) due to the intractable integration inside of nonlinear function $f$. However, we can exploit the Fenchel duality trick (Nguyen et al., 2010; Dai et al., 2017; Nachum & Dai, 2020) to (87), which leads to the adversarial spectral representation learning,

$$\min_{\mathbb{P}(z|x)} \mathbb{E}_{x,x'}\mathbb{E}_{\mathbb{P}(z|x')}\left[f\left(\frac{\mathbb{P}\left(z|x\right)}{\int \mathbb{P}\left(z|x'\right)\mathbb{P}\left(x'|x\right)dx'}\right)\right] \tag{101}$$

$$= \min_{\mathbb{P}(z|x)} \mathbb{E}_{x,x'}\mathbb{E}_{\mathbb{P}(z|x')}\left[\max_{g_{x,z}} \frac{\mathbb{P}\left(z|x\right)}{\int \mathbb{P}\left(z|x'\right)\mathbb{P}\left(x'|x\right)dx'} \cdot g_{x,z} - f^*\left(g_{x,z}\right)\right] \tag{102}$$

$$= \min_{\mathbb{P}(z|x)} \max_{g(x,z)} \mathbb{E}_{x\sim\mathcal{D}}\mathbb{E}_{\mathbb{P}(z|x)}\left[g\left(x,z\right)\right] - \mathbb{E}_{x\sim\mathcal{D}}\mathbb{E}_{\mathbb{P}(z|x')\mathbb{P}(x'|x)}\left[f^*\left(g\left(x,z\right)\right)\right]. \tag{103}$$

Particularly, for continuous $z$, we can exploit the reparametrization trick for gradient estimation; while for discrete $z$, *e.g.*, softmax parametrization (78), we can enumerate for the expectation and gradient calculation. Although (101) is a valid estimator for $\mathbb{P}\left(z|x\right)$, it is a min-max saddle point optimization, which might need more extra strategies for learning stability in practice.

## 6 Nonlinear Spectral Representation

In Section 3, we consider the *linear sufficient* representation from the factorization perspective. With the energy-based spectral representation discussed in Section 4, we observe that the nonlinear sufficient representation has a corresponding linear sufficient representation but with *infinite dimensions*. Both the linear and nonlinear representations are sufficient, in the sense that the downstream expectation can be fully expressed based on the representation. However, the major difference lies in the downstream formulations for representing $\mathbb{P}\left(y|x\right)$ and $\mathbb{E}\left[y|x\right]$ as we revealed in corresponding section.

These observations imply that although both linear spectral representation and energy-based spectral representation are sufficient for downstream task, there exists a tradeoff between the downstream task parametrization and dimensionality of the sufficient representation. Concretely, with linear factorization, linear models are enough for the downstream tasks, but requires low-rank assumption on $\varphi\left(x\right)$ to represent $\mathbb{P}\left(x'|x\right)$; while for nonlinear factorization, a compact representation $\upsilon\left(x\right)$ will induce infinite-dimensional linear spectral representation $\varphi_\omega\left(x\right)$ without low-rank assumption, however, it requires the flexible model, *e.g.*, neural networks and kernel machine, for downstream tasks.

This inspires us to generalize nonlinear sufficient representation beyond $\exp\left(\cdot\right)$ in energy-based models, which also justifies the usage of "projector" in most of the representation learning algorithms. Concretely, following the equivalent linear sufficient representation for energy-based spectral representation by random feature in kernel view, we can simply replace the $\exp\left(\cdot\right)$ with arbitrary positive-definite kernel, *i.e.*,

$$\mathbb{P}\left(x'|x\right) = \mathbb{P}\left(x'\right)k\left(\upsilon\left(x\right), \upsilon\left(x'\right)\right), \tag{104}$$

which generalizes the energy-based spectral representation (48) and (52). The parametrization (104) admits a spectral decomposition $\varphi\left(x\right) = f\left(\upsilon\left(x\right)\right)$. Although $f\left(\cdot\right)$ is unknown, it still implies the nonlinear sufficient representation $\upsilon\left(x\right)$ with a corresponding linear spectral representation $f\left(\upsilon\left(x\right)\right)$. So we can parametrize the nonlinear representation $\upsilon\left(x\right)$ with the projector $f\left(\cdot\right) = \mathtt{nn}_\theta\left(\cdot\right)$, leading to

$$\mathbb{P}\left(x'|x\right) = \mathbb{P}\left(x'\right)\left\langle\mathtt{nn}_\theta\left(\upsilon\left(x\right)\right), \mathtt{nn}_\theta\left(\upsilon\left(x'\right)\right)\right\rangle, \tag{105}$$

which will be learned together by exploiting the objectives we discussed in Section 3, 4, and 5.

Obviously, once we have the nonlinear spectral representation and its corresponding projector, we obtain the formulation of $\mathbb{E}[y|x]$ linearly w.r.t. $\varphi(x)$, *i.e.*,

$$\mathbb{E}[y|x] = w^\top \varphi(x) = w^\top \mathbf{nn}_\theta(\upsilon(x)). \tag{106}$$

We can further generalize the kernel-based nonlinear representation in two ways:

- **Asymmetric Projector.** Instead of considering the symmetric decomposition (5), we can further nonlinearize the asymmetric decomposition (3). Concretely, recall the decomposition used in BYOL, *i.e.*,

$$\mathbb{P}(x'|x) = \mathbb{P}(x)\langle \xi(x'), \Lambda\xi(x)\rangle, \tag{107}$$

with $\Lambda$ and $\xi$ defined in (19), we introduce nonlinear parametrization to generalize $\Lambda\xi(x)$, leading to

$$\mathbb{P}(x'|x) = \mathbb{P}(x')\langle \upsilon(x'), \mathbf{nn}_\theta(\upsilon(x))\rangle. \tag{108}$$

The asymmetric projector has been exploited in several existing algorithm (Chen & He, 2021). In fact, the asymmetric project (108) can be view as a specialization of multi-modal spectral representation parametrization, which will be explained in Section 7.

- **Neural Product.** Instead of the product of neural networks in (105), we consider the neural networks on product, *i.e.*,

$$\mathbb{P}(x'|x) = \mathbb{P}(x')\mathbf{nn}_\theta(\langle \upsilon(x'), \upsilon(x)\rangle). \tag{109}$$

**Remark (Flexibility vs. Predictor Parametrization)** As a payoff to the flexibility of nonlinear generalization in (108) and (109), we may lose the ability to get an explicit closed-form parametrization of $\mathbb{P}(y|x)$ or $\mathbb{E}[y|x]$. Therefore, we must use neural network parametrization upon $\upsilon(x)$ for downstream task predictors. However, the flexibility brings better modeling for $\mathbb{P}(x'|x)$, leading to better generation ability, which might be more important and desirable.

## 6.1 Nonlinear Spectral Representation Learning

With the nonlinear spectral representation parametrization proposed, we can recover several existing representation learning algorithm, therefore, provide more understanding.

- **Simple Siamese Representation Learning (SimSiam) (Chen & He, 2021).** SimSiam (Chen & He, 2021) can be understood as nonlinear extension to BYOL (Grill et al., 2020). Concretely, as we derived in (31) and (32), for $t$-th update, BYOL is iterating

$$\min_{\xi,\Lambda} \ \mathbb{E}_{\mathbb{P}(x',x)}\left[\|\Lambda\xi(x) - \xi_t(x')\|_2^2\right], \tag{110}$$

to implement power iteration for spectral decomposition (107). In the nonlinear extension (108), we replace $\Lambda\xi(x)$ in (107) with $\mathbf{nn}_\theta(\nu(x))$. Plug this into (110), we obtain the $t$-th iteration optimization

$$\min_{\upsilon,\mathbf{nn}_\theta} \ \mathbb{E}_{\mathbb{P}(x',x)}\left[\|\mathbf{nn}_\theta(\upsilon(x)) - \upsilon_t(x')\|_2^2\right]$$
$$= \mathbb{E}\left[\|\mathbf{nn}_\theta(\upsilon(x))\|_2^2\right] + \mathbb{E}\left[\|\upsilon_t(x')\|_2^2\right] - 2\mathbb{E}_{\mathbb{P}(x',x)}\left[\mathbf{nn}_\theta(\upsilon(x))^\top \upsilon(x')\right]. \tag{111}$$

Additionally, with normalization condition, *i.e.*, $\|\mathbf{nn}_\theta(\upsilon(x))\|_2^2 = 1$ and $\|\upsilon_t(x')\|_2^2 = 1$, we obtain an equivalent objective as

$$\min_{\upsilon,\mathbf{nn}_\theta} \ -\mathbb{E}_{\mathbb{P}(x',x)}\left[\mathbf{nn}_\theta(\upsilon(x))^\top \upsilon(x')\right]. \tag{112}$$

In practice, the normalization condition is implemented by self-normalization, we achieve the SimSiam objective, *i.e.*,

$$\min_{\upsilon,\mathbf{nn}_\theta} \ -\mathbb{E}_{\mathbb{P}(x',x)}\left[\left(\frac{\mathbf{nn}_\theta(\upsilon(x))}{\|\mathbf{nn}_\theta(\upsilon(x))\|_2}\right)^\top \frac{\upsilon_t(x')}{\|\upsilon_t(x')\|_2}\right]. \tag{113}$$

## 6.2 More Nonlinear Spectral Representation Learning

As we can see, SimSiam exploits the variational fixed-point iteration for nonlinear spectral representation for (108), which can also be applied for (104), (105), and (109).

We can extend the other algorithms we explained in Section 3.2 for nonlinear spectral representation learning. For demonstration, we instantiate one method for nonlinear parametrization, but we emphasize that other algorithms can potentially also be applied.

- **Unnormalized Distribution Fitting.** The classification-based NCE and ranked-based NCE can be directly applied for nonlinear spectral representation (104). By plugging the kernel based nonlinear generation (104) into (37) and (39), we obtain the corresponding NCE objectives, *i.e.*,

$$\max_{v,k} \ \mathbb{E}_{\mathbb{P}(x,x')} \left[ \log \frac{k\left(v\left(x\right), v\left(x'\right)\right)}{1 + k\left(v\left(x\right), v\left(x'\right)\right)} \right] - \mathbb{E}_{\mathbb{P}(x)\mathbb{P}(x')} \left[ \log\left(1 + k\left(v\left(x\right), v\left(x'\right)\right)\right) \right], \tag{114}$$

$$\max_{v,k} \ \mathbb{E}_{\mathbb{P}(x,x')} \left[ \log k\left(v\left(x\right), v\left(x'\right)\right) \right] - \mathbb{E}_{\mathbb{P}(x)} \left[ \log \sum_{x' \sim \mathbb{P}(x')} k\left(v\left(x\right), v\left(x'\right)\right) \right]. \tag{115}$$

The algorithm is also applicable to the parametrization (108) and (109).

- **Density Ratio Fitting.** Recall the spectral representation learning via $f$-divergence in Section 3.2, we can plug the nonlinear spectral representation (104) to (42), which leads to the learning objective,

$$\max_{v,k} \ \mathbb{E}_{\mathbb{P}(x',x)} \left[ f'\left(k\left(\langle v\left(x\right), v\left(x'\right)\rangle\right)\right) \right] - \mathbb{E}_{\mathbb{P}(x)\mathbb{P}(x')} \left[ f^*\left(f'\left(k\left(\langle v\left(x\right), v\left(x'\right)\rangle\right)\right)\right) \right]. \tag{116}$$

The algorithm is also applicable to the parametrization (108) and (109).

# 7 Multi-Modal Spectral Representation

In this section, we broaden our spectral representation view for multi-modal representation. Specifically, we consider the parametrization for singular value decomposition to pointwise mutual information, *i.e.*,

$$\frac{\mathbb{P}\left(x,y\right)}{\mathbb{P}\left(x\right)\mathbb{P}\left(y\right)} = \langle \varphi\left(x\right), \mu\left(y\right)\rangle, \tag{117}$$

which implies

$$\mathbb{P}\left(x|y\right) = \mathbb{P}\left(x\right)\langle \varphi\left(x\right), \mu\left(y\right)\rangle, \quad \text{and} \quad \mathbb{P}\left(y|x\right) = \mathbb{P}\left(y\right)\langle \varphi\left(x\right), \mu\left(y\right)\rangle. \tag{118}$$

As we explained in Section 2 and Section 9.1, the representation is sufficient for representing regressor $\mathbb{E}\left[y|x\right]$ in (2) and classifier.

Similarly, we can also extend the linear multi-modal representation for energy-based ((48) and (52)), latent-variable ((76)), and nonlinear spectral representation ((104), and (109)), and modify the corresponding learning methods for the parametrization, respectively. We mainly focus on the energy-based multi-modal representation as a demonstration. Concretely, we consider the EBM

$$\frac{\mathbb{P}\left(x,y\right)}{\mathbb{P}\left(x\right)\mathbb{P}\left(y\right)} \propto \exp\left(\langle \phi\left(x\right), \nu\left(y\right)\rangle\right), \tag{119}$$

which leads to the corresponding distributions

$$\mathbb{P}\left(x|y\right) = \frac{1}{Z\left(y\right)}\mathbb{P}\left(x\right)\exp\left(\langle \phi\left(x\right), \nu\left(y\right)\rangle\right), \quad \mathbb{P}\left(y|x\right) = \frac{1}{Z\left(x\right)}\mathbb{P}\left(y\right)\exp\left(\langle \phi\left(x\right), \nu\left(y\right)\rangle\right), \tag{120}$$

$$\mathbb{P}\left(x,y\right) = \frac{1}{Z}\mathbb{P}\left(x\right)\mathbb{P}\left(y\right)\exp\left(\langle \phi\left(x\right), \nu\left(y\right)\rangle\right), \tag{121}$$

with

$$Z\left(y\right) = \int \mathbb{P}\left(x\right)\exp\left(\left\langle\phi\left(x\right),\nu\left(y\right)\right\rangle\right)dx, \quad Z\left(x\right) = \int \mathbb{P}\left(y\right)\exp\left(\left\langle\phi\left(x\right),\nu\left(y\right)\right\rangle\right)dy, \quad (122)$$

$$Z = \int \mathbb{P}\left(x\right)\mathbb{P}\left(y\right)\exp\left(\left\langle\phi\left(x\right),\nu\left(y\right)\right\rangle\right)dxdy. \quad (123)$$

### 7.1 Energy-based Multi-Modal Spectral Representation Learning

With the energy-based parametrization (119), we can recast several representative multi-modal representation learning from spectral view.

- **CLIP (Radford et al., 2021) and SigLIP (Zhai et al., 2023).** We apply ranking-based NCE for estimating $\mathbb{P}\left(y|x\right)$ and $\mathbb{P}\left(x|y\right)$ in (120), leading to

$$\min_{\phi,\nu} \ -\mathbb{E}_{\mathbb{P}(x,y)}\left[\phi\left(x\right)^{\top}\nu\left(y\right)\right] + \mathbb{E}_{\mathbb{P}(x)}\left[\log\sum_{i=1}\exp\left(\phi\left(x\right)^{\top}\nu\left(y_i\right)\right)\right]$$
$$-\mathbb{E}_{\mathbb{P}(x,y)}\left[\phi\left(x\right)^{\top}\nu\left(y\right)\right] + \mathbb{E}_{\mathbb{P}(y)}\left[\log\sum_{i=1}\exp\left(\phi\left(x_i\right)^{\top}\nu\left(y\right)\right)\right], \quad (124)$$

  which is exactly the objective of Contrastive Language-Image Pre-training (CLIP) (Radford et al., 2021). We apply classification-based NCE for estimating $\mathbb{P}\left(y|x\right)$ in (120), leading to

$$\min_{\phi,\nu} \ -\mathbb{E}_{\mathbb{P}(x,y)}\left[\log\frac{1}{1+\exp\left(-\phi\left(x\right)^{\top}\nu\left(y\right)\right)}\right] \quad - \quad \mathbb{E}_{\mathbb{P}(x)\mathbb{P}(y)}\left[\log\frac{1}{1+\exp\left(\phi\left(x\right)^{\top}\nu\left(y\right)\right)}\right]. \quad (125)$$

  which is exactly the objective of Sigmoid Language-Image Pre-training (SigLIP) (Zhai et al., 2023). Similarly, we can also apply classification-based NCE for estimating $\mathbb{P}\left(x|y\right)$ and $\mathbb{P}\left(x,y\right)$ in (120), which leads to the same objective to (125).

  As we can see from the difference between (124) and (125), the SigLIP is compatible with stochastic gradient with unbiased gradient estimator, while CLIP requires large batchsize due to the log-sum-exp operator. However, empirical study shows that CLIP works better than SigLIP, which is aligned with the theoretical justification in (Ma & Collins, 2018) about the consistency condition: the solution from classification-based NCE is consistent with the model with normalization condition that $\int\exp\left(\phi\left(x\right)^{\top}\nu\left(y\right)\right)dxdy = 1$, $\int\exp\left(\phi\left(x\right)^{\top}\nu\left(y\right)\right)dy = 1$, and $\int\exp\left(\phi\left(x\right)^{\top}\nu\left(y\right)\right)dx = 1$. Therefore, the SigLIP empirically introduces an additional temperature parameter, *i.e.*, $\exp\left(\frac{\phi(x)^{\top}\nu(y)}{\tau}\right)$, requiring heaving tuning.

- **SogLIP (Yuan et al., 2022), AmorLIP (Sun et al., 2025), and NeuCLIP (Wei et al., 2025).** To bypass the implicit assumption in SigLIP while also reduce the large-batch size issue in CLIP, there have been a few attempts, in which SogLIP (Yuan et al., 2022), AmorLIP (Sun et al., 2025), and NeuCLIP (Wei et al., 2025) are the representative. These algorithms can all be viewed as approximations of maximum log-likelihood estimation of (120).

  The MLE of $\mathbb{P}\left(x|y\right)$ and $\mathbb{P}\left(y|x\right)$ is

$$\max_{\phi,\nu} \ 2\mathbb{E}_{\mathbb{P}(x,y)}\left[\phi\left(x\right)^{\top}\nu\left(y\right)\right] - \mathbb{E}_{\mathbb{P}(x)}\left[\log Z\left(x\right)\right] - \mathbb{E}_{\mathbb{P}(y)}\left[\log Z\left(y\right)\right], \quad (126)$$

  with the same gradient as the objective

$$\max_{\phi,\nu} \ 2\mathbb{E}_{\mathbb{P}(x,y)}\left[\phi\left(x\right)^{\top}\nu\left(y\right)\right] - \mathbb{E}_{\mathbb{P}(x)\mathbb{P}(y)}\left[\frac{\exp\left(\phi\left(x\right)^{\top}\nu\left(y\right)\right)}{\mathtt{stopgrad}\left(Z\left(x\right)\right)}\right] - \mathbb{E}_{\mathbb{P}(x)\mathbb{P}(y)}\left[\frac{\exp\left(\phi\left(x\right)^{\top}\nu\left(y\right)\right)}{\mathtt{stopgrad}\left(Z\left(y\right)\right)}\right]. \quad (127)$$

The major difficulty in estimating the gradients to (127) is the intractability of $Z(x)$ and $Z(y)$. These algorithms are exploiting different ways for the partition function approximation. Particularly, in SogLIP (Yang et al., 2023), nonparametric function is exploited for partition functions: for each sample $x$ and $y$, the corresponding $Z_x$ and $Z_y$ are scalars and estimated by moving averaging, *i.e.*,

$$Z_x^t = (1 - \eta) Z_x^{t-1} + \eta \sum_{y \in B} \exp\left(\phi(x)^\top \nu(y)\right), \tag{128}$$

$$Z_y^t = (1 - \eta) Z_y^{t-1} + \eta \sum_{x \in B} \exp\left(\phi(x)^\top \nu(y)\right). \tag{129}$$

The nonparametric parametrization of $Z(x)$ and $Z(y)$ induces easy way for updating, but $O(n)$ memory cost and additional overhead in function computing through retrieval, which is unaffordable for large datasets. AmorLIP (Sun et al., 2025) bypasses this by exploiting the property that $Z(x)$ and $Z(y)$ are also representable by $\phi(x)$ and $\nu(y)$ as we proved in (53). Then, Sun et al. (2025) introduces several variational objectives to fit a simple model upon the learned representation $\phi(x)$ and $\nu(y)$ for approximating $Z(x)$ and $Z(y)$, respectively. Following the idea, NeuCLIP (Wei et al., 2025) introduces another objective for fitting partition funcitons by exploiting Fenchel dual trick (Dai et al., 2017; 2019; Nachum & Dai, 2020).

## 7.2 More Multi-Modal Spectral Representation Learning

In fact, besides NCE and variational MLE for energy-based multi-modal spectral representation, the algorithms we introduced in Section 3, Section 4 and Section 5, *e.g.*, fixed-point iteration, density ratio estimation, variational ELBO and adversarial training, and so on, can be tailored for multi-modal spectral representation learning with asymmetric parametrization (117) and (119). The derivation of these variants are similar and we will not repeat. Here we elaborate the diffusion loss for learning multi-modal energy-based spectral representation (119).

Following the derivation in Section 4.1, where we connect the diffusion and energy-based model (Shribak et al., 2024), we augment our energy-based parametrization (119) with noise level, *i.e.*,

$$\mathbb{P}(\tilde{x}, \tilde{y}; \beta_1, \beta_2) \propto \int \mathbb{P}(\tilde{x}|x, \beta_1) \mathbb{P}(x) \mathbb{P}(\tilde{y}|y, \beta_2) \mathbb{P}(y) \exp\left(\langle \phi(x), \nu(y) \rangle\right) dx dy \tag{130}$$

$$= \int \mathbb{P}(\tilde{x}|x, \beta_1) \mathbb{P}(x) \mathbb{P}(\tilde{y}|y, \beta_2) \mathbb{P}(y) \langle \varphi_\omega(x), \mu_\omega(y) \rangle_{\mathcal{N}(\omega)} dx dy \tag{131}$$

$$= \left\langle \underbrace{\int \mathbb{P}(\tilde{x}|x, \beta_1) \mathbb{P}(x) \varphi_\omega(x) dx}_{\varphi_\omega(\tilde{x}; \beta_1)}, \underbrace{\int \mathbb{P}(\tilde{y}|y, \beta_2) \mathbb{P}(y) \mu_\omega(y) dy}_{\mu_\omega(\tilde{y}; \beta_2)} \right\rangle_{\mathcal{N}(\omega)} \tag{132}$$

$$= \exp\left(\langle \tilde{\phi}(\tilde{x}; \beta_1), \tilde{\nu}(\tilde{y}; \beta_2) \rangle\right). \tag{133}$$

Then, we consider the diffusion loss to learn $\tilde{\phi}(\tilde{x}; \beta_1)$ and $\tilde{\nu}(\tilde{y}; \beta_2)$ based on the Tweedie's Identity for $\mathbb{P}(\tilde{x}|y; \beta)$ and $\mathbb{P}(\tilde{y}|x; \beta)$, which leads to the diffusion loss

$$\min_{\tilde{\phi}, \tilde{\nu}} \mathbb{E}_{x, y \sim \mathbb{P}(x,y), \tilde{x}, \tilde{y}, \beta_1, \beta_2} \left[ \left\| \tilde{x} + \beta_1 \sigma^2 \nabla_{\tilde{x}} \phi(\tilde{x}; \beta_1)^\top \tilde{\nu}(\tilde{y}; \beta_2) - \sqrt{1 - \beta_1} x \right\|^2 \right] \tag{134}$$

$$+ \mathbb{E}_{x, y \sim \mathbb{P}(x,y), \tilde{x}, \tilde{y}, \beta_1, \beta_2} \left[ \left\| \tilde{y} + \beta_2 \sigma^2 \phi(\tilde{x}; \beta_1)^\top \nabla_{\tilde{y}} \tilde{\nu}(\tilde{y}; \beta_2) - \sqrt{1 - \beta_2} y \right\|^2 \right]. \tag{135}$$

**Remark (Discrete Diffusion):** In the diffusion loss (134), we implicitly assume the gradient exists, which is not satisfied for discrete variables. We can simply change the loss to discrete diffusion loss. Concretely, we denote $x$ as the text, *i.e.*, discrete variables, then, we keep the second term in (134) and only replace the first term with masked diffusion loss (Shi et al., 2024), *i.e.*,

$$\mathbb{E}_{x, y \sim \mathbb{P}(x,y), \tilde{x}, \tilde{y}, \beta_1, \beta_2} \left[ -x^\top \phi(\tilde{x}; \beta_1)^\top \tilde{\nu}(\tilde{y}, \beta_2) \right]. \tag{136}$$

In fact, other diffusion loss for discrete variables can also be exploited (Austin et al., 2021; Sun et al., 2022).

## 8 Spectral Representation in History

In history, there have been a rich literature in exploiting the spectral representation. Below we establish the connections of the existing SSL algorithms to corresponding dimension reduction/embedding algorithms in statistics and manifold learning from spectral representation view.

- **Stochastic Neighbor Embedding and Neighborhood Component Analysis.** We remark that the stochastic neighbor embedding (SNE) (Hinton, 2002) and neighborhood component analysis (NCA) (Goldberger et al., 2004) can be recovered from NCE (60) with linear or nonparametric form of $\upsilon(x)$ in (48).

  Specifically, plugging $\upsilon(x) = \frac{Wx}{\|Wx\|}$ into (48), we have the energy-based model,

  $$\mathbb{P}(x'|x) = \mathbb{P}(x') \exp\left(\frac{Wx}{\|Wx\|}^\top \frac{Wx'}{\|Wx'\|}\right), \tag{137}$$

  with corresponding NCE as

  $$\min_{W} \ -\mathbb{E}_{\mathbb{P}(x,x')}\left[\frac{Wx}{\|Wx\|}^\top \frac{Wx'}{\|Wx'\|}\right] + \mathbb{E}_{\mathbb{P}(x)}\left[\log\sum_{i=1}\exp\left(\frac{Wx}{\|Wx\|}^\top \frac{Wx'_i}{\|Wx'_i\|}\right)\right]. \tag{138}$$

  It recovers the log loss of NCA. If we consider the nonparametrization form of $\upsilon(x_i) = y_i$ for each sample $x_i$ in $\mathcal{D}$, the SNE (Hinton & Roweis, 2002) is recovered. Similarly, t-SNE (Maaten & Hinton, 2008) can be recovered by considering $t$-distribution, i.e., $\mathbb{P}(x'|x) = \mathbb{P}(x')\left(3 - 2\frac{Wx}{\|Wx\|}^\top \frac{Wx'}{\|Wx'\|}\right)^{-1}$.

- **Locality Preserving Projections and Laplacian Embedding.** Recall the condition (6)

  $$\frac{\mathbb{P}(x',x)}{\mathbb{P}(x)\mathbb{P}(x')} = \langle\varphi(x'),\varphi(x)\rangle \Rightarrow \frac{\mathbb{P}(x|x')}{\mathbb{P}(x)} = \langle\varphi(x'),\varphi(x)\rangle, \tag{139}$$

  which implies $\varphi(x)$ is the eigenfunction of $\frac{\mathbb{P}(x|x')}{\mathbb{P}(x)}$. Locality Preserving Projections and Laplacian Embedding can be derived to implement this decomposition with linear or nonparametric form with empirical Monte Carlo approximation to $\frac{\mathbb{P}(x|x')}{\mathbb{P}(x)} = \frac{\delta_{x',\epsilon}(x)\exp\left(-\|x-x'\|^2/2\right)}{Z(x')}$, $Z(x') = \int\mathbb{P}(x)\delta_{x',\epsilon}(x)\exp\left(-\|x-x'\|^2/2\right)dx$, $\delta_{x',\epsilon}(x) = \begin{cases}1 & \text{if } \|x_i - x_j\| \leqslant \epsilon \\ 0 & \text{otherwise}\end{cases}$.

  Concretely, given dataset $\mathcal{D}$, we exploit $W_{ij} = \begin{cases}\exp\left(-\frac{\|x_i-x_j\|}{2}\right) & \text{if } \|x_i-x_j\| \leqslant \epsilon \\ 0 & \text{otherwise}\end{cases}$ and $D = \text{diag}(W\mathbf{1})$ to approximate $Z(x')$. Then, we have the eigendecomposition condition

  $$\int\frac{\mathbb{P}(x|x')}{\mathbb{P}(x)}\varphi(x)\mathbb{P}(x)\,dx = \Lambda\varphi(x') \approx D^{-1}WY = n\Lambda Y \tag{140}$$

  with $\varphi(x_i) = y_i$. This recovers the Laplacian Embedding (Belkin & Niyogi, 2001).

  If we exploit the linear parametrization $\varphi(x) = Vx \in \mathbb{R}^{d\times 1}$ to (140), replacing $Y$, we obtain the condition

  $$D^{-1}WXV^\top = n\Lambda XV^\top,$$

  which implies the optimization

  $$\max_{V} \ \text{tr}\left(VX^\top D^{-1}WXV^\top\right). \tag{141}$$

  Usually, to avoid trivial solution, we add the orthonormal constraint on $\varphi(x)$, i.e., $\mathbb{E}_{\mathbb{P}(x)}\left[Vxx^\top V^\top\right] = \mathbf{I}_d$. It recovers the Locality Preserving Projections (He & Niyogi, 2003).

- **Multidimensional Scaling, PCA and Isomap.** The principal component analysis (PCA) (Pearson, 1901), multidimensional scaling (MDS) (Kruskal, 1964), and Isomap (Tenenbaum et al., 2000) can be recast to the same optimization with different estimation of $\frac{\mathbb{P}(x,x')}{\mathbb{P}(x)\mathbb{P}(x')}$.

Concretely, we start with the variaitonal characteristic property of spectral decompsition (16), *i.e.*,

$$\varphi(x) = \underset{\xi(x) \text{ is } d\text{-rank}}{\operatorname{argmax}} \underbrace{\int \frac{\mathbb{P}(x, x')}{\mathbb{P}(x)\mathbb{P}(x')}\mathbb{P}(x)\xi(x)^\top \mathbb{P}(x')\xi(x')\, dxdx'}_{\mathbb{E}_{\mathbb{P}(x)\mathbb{P}(x')}\left[\frac{\mathbb{P}(x,x')}{\mathbb{P}(x)\mathbb{P}(x')}\xi(x)^\top \xi(x')\right]}. \tag{142}$$

Since all the $d$-rank basis are equivalent in terms of representation ability, in (142), we do not enforce for a particular basis. It will be instantiated in different algorithms below.

MDS (Kruskal, 1964) can be instantiated from (142) as an U-statistics upon sample $\mathcal{D}$ with nonparametric $\xi(x)$ with additional constraints. Concretely, given $\mathcal{D} = \{x_i\}_{i=1}^n$, we set $k(x, x') = \frac{\mathbb{P}(x,x')}{\mathbb{P}(x)\mathbb{P}(x')} = -\|x - x'\|^2$, and consider the nonparametric $\xi(x_i) = z_i$ with centering constraint, *i.e.*, $\sum_{i=1} z_i = 0$. To ensure the centering constraint, we consider the reparametrization $Z = HY \in \mathbb{R}^{n\times d}$, where $H = \mathbf{I} - \frac{1}{n}\mathbf{1}\mathbf{1}^\top$. Obviously, it is easy to verify that

$$\sum_{i=1}^n z_i = \mathbf{1}^\top Z = \mathbf{1}^\top Y - \frac{1}{n}\left(\mathbf{1}^\top \mathbf{1}\right)\mathbf{1}^\top Y = \mathbf{1}^\top Y - \mathbf{1}^\top Y = 0. \tag{143}$$

Meanwhile, with the parametrization $Z = \left(\mathbf{I} - \frac{1}{n}\mathbf{1}\mathbf{1}^\top\right)Y$, full-rank constraint can also be easily implemented by enforcing $Y^\top Y = \mathbf{I}$. With this parametrization of $\xi(x)$, we obtain the optimization,

$$\max_{Y^\top Y = nI_d} \sum_{i,j=1}^n -\|x_i - x_j\|^2 z_i^\top z_j = \operatorname{tr}\left(Y^\top HKHY\right), \tag{144}$$

which is exactly MDS.

Isomap can be naturally extended from MDS with different kernel estimation for $\frac{\mathbb{P}(x,x')}{\mathbb{P}(x)\mathbb{P}(x')}$ and parametrization of $\xi(x)$. Particularly, in Isomap (Tenenbaum et al., 2000), the Euclidean kernel $k(x, x')$ is replaced by geodesic distance, which is computed based on shortest paths over given dataset.

Obviously, the optimization (144) is exactly the same as kernel PCA (Schölkopf et al., 1998), which implies the MDS is a special case of kernel PCA with special kernel (Williams, 2000). Particularly, with linear kernel and linear parametrization $Y = XW$ in (144), we recover the classic PCA. For simplicity, we assume $X$ is centeralized, *i.e.*, $HX = X$. Plug the linear kernel into (144), the KKT condition implies,

$$XX^\top Y = Y\Lambda \Rightarrow \cancel{X^\top X}X^\top XW = \cancel{X^\top X}W\Lambda. \tag{145}$$

With the constraint $W^\top X^\top XW = I$ and KKT condition, we achieve the solution that $W = U\Lambda^{-\frac{1}{2}}$, where $U$ and $\Lambda$ denotes the eigenvector and eigenvalue, respectively, *i.e.*, $X^\top X = U^\top \Lambda U$.

- **Canonical Component Analysis (CCA).** The canonical component analysis (CCA) (Hotelling, 1936) can be recovered from (117) for spectral representation in multi-view setting. We start with kernel CCA and reduce to linear CCA as special case. Following the definition of CCA (Bach & Jordan, 2002; Fukumizu et al., 2007; Michaeli et al., 2015), which is finding the transformation to maximize the correlation, the objective of CCA can be written as

$$\max_{\varphi(\cdot),\mu(\cdot)} \operatorname{cov}[\varphi,\mu] := \mathbb{E}_{\mathbb{P}(x,y)}\left[\varphi(x)^\top \mu(y)\right] = \mathbb{E}_{\mathbb{P}(x)\mathbb{P}(y)}\left[\frac{\mathbb{P}(x,y)}{\mathbb{P}(x)\mathbb{P}(y)}\varphi(x)^\top \mu(y)\right] \tag{146}$$

$$\text{s.t. } \mathbb{E}_{\mathbb{P}(x)}\left[\varphi(x)\varphi(x)^\top\right] = \mathbf{I}, \quad \mathbb{E}_{\mathbb{P}(y)}\left[\mu(y)\mu(y)^\top\right] = \mathbf{I}, \tag{147}$$

which is exactly the variational characteristic property of singular value decomposition of (117). Plug the linear parametrization, $\varphi(x) = Wx$ and $\mu(y) = Vy$, we recover classic CCA (Hotelling, 1936).

# 9 Application of Spectral Representation

As we discussed in Section 2, the learned representation is sufficient for downstream regression task. In this section, we discuss the the usage of the sufficient representation for other tasks, including causal inference with unobserved confounder (Sun et al., 2024), control (Ren et al., 2025) and reinforcement learning (Gao et al., 2025), animal behavior research (Wang et al., 2025), and controllable synthesis in foundation models.

As we reveal, for nonlinear sufficient representation, there exists a corresponding linear sufficient representation. Therefore, for simplicity, we take the sufficient linear representation as an example. The same argument can be generalized for applying energy-based and latent variable spectral representation for downstream tasks.

## 9.1 Regression and Classification

The motivation for linear spectral sufficient representation comes from linear representing all the optimal regressors to downstream least square regression in (2), *i.e.*,

$$\mathbb{E}\left[y|x\right] = \int y \mathbb{P}\left(y|x\right) dy = \left\langle \varphi\left(x\right), \underbrace{\int y \mu\left(y\right) dy}_{w} \right\rangle. \tag{148}$$

The same idea can be generalized for classification. Recall the decision rule, which minimizes the conditional risk, *i.e.*,

$$R\left(y=i|x\right) = \sum_{j=1}^{k} \omega\left(y=i, y=j\right) \mathbb{P}\left(y=j|x\right) \Rightarrow R\left(\cdot|x\right) = \left\langle \varphi\left(x\right), \underbrace{\sum_{j=1}^{k} \omega\left(\cdot, y=j\right) \mu\left(y=j\right)}_{W:=\Omega\mu(\cdot)\in\mathbb{R}^{d\times k}} \right\rangle. \tag{149}$$

It implies that the conditional risk can be linearly represented by $\phi\left(x\right)$, and the classification task can be completed by

$$\underset{y\in\{1,\ldots,k\}}{\operatorname{argmin}} \ \langle W_y, \varphi\left(x\right)\rangle. \tag{150}$$

Particularly, if we set $\omega\left(i,j\right) = \begin{cases} 0, & \text{if} \quad i = j \\ 1, & \text{if} \quad i \neq j \end{cases}$, the conditional risk reduces to posterior, which recovers the Bayes decision rule, *i.e.*,

$$\underset{y\in\{1,\ldots,k\}}{\operatorname{argmin}} \ \mathbb{P}\left(y|x\right) = \langle W_y, \varphi\left(x\right)\rangle. \tag{151}$$

As we can linearly representation the posterior $\mathbb{P}\left(y|x\right)$ by $\varphi\left(x\right)$, it is obviously the related statistics, *e.g.*, mean, median, moments, and characteristic function, can be represented as well. Therefore, the function family linearly/nonlinearly-composed by sufficient representation can be used for any loss function whose solution is some statistics w.r.t. $\mathbb{P}\left(y|x\right)$, *e.g.*, logistic/softmax classification, robust $L1$-loss for regression, and so on.

## 9.2 Causal Inference with Hidden Confounders

The spectral representation has been exploited in (Sun et al., 2024) for function identification in causal inference with hidden confounder, including instrumental variable (IV) regression (Figure 1a and Figure 1b), and proxy causal inference (Figure 1c).

We take the IV regression as the example for illustration. We define the conditional expectation operator $E : L_2\left(P_x\right) \to L_2\left(P_z\right)$ as $Ef = \mathbb{E}\left[f\left(x\right)|z\right]$ for any $f \in L_2\left(P_x\right)$. We omit the rigorous assumptions, but only reveal the spectral representation structure.

We follow the notations in Figure 1a. Specifically, we denote the factorization

$$\mathbb{P}\left(x|z\right) = \mathbb{P}\left(x\right) \langle \varphi\left(x\right), \mu\left(z\right)\rangle.$$

Then, Dai et al. (2017) recast the IV problem as a saddle-point problem by exploiting Fenchel duality, *i.e.*,

$$\min_{f} \; \mathbb{E}_z \left[ \left( \mathbb{E}_{x,y|z} \left[ y - f(x) \right] \right)^2 \right] + \lambda \Omega(f) \tag{152}$$

$$\Rightarrow \min_{f} \max_{g} \; \mathbb{E}_{x,z,y} \left[ \left( y - f(x) \right) g(z) - \frac{1}{2} g(z)^2 \right] + \lambda \Omega(f), \tag{153}$$

where $\lambda > 0$, $\Omega(\cdot)$ denotes some regularization,

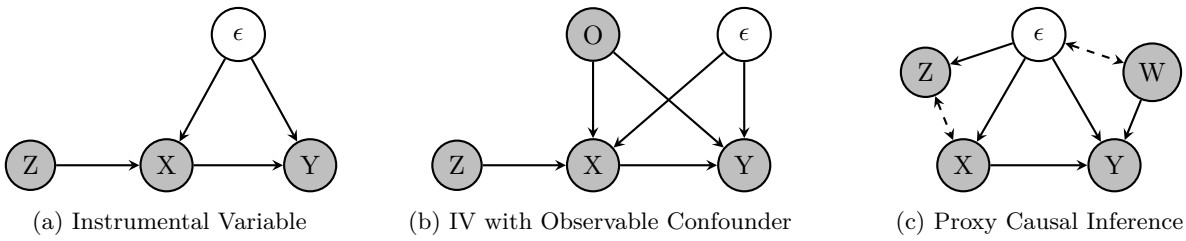

|  (a) Instrumental Variable | (b) IV with Observable Confounder | (c) Proxy Causal Inference |

Figure 1: Causal graphs considered, with grey node denoting the observable variable.

Sun et al. (2024) reveal the fact that both $f$ and $g$ can be represented by the primal-dual spectral representation of $\mathbb{P}(x|z)$. Specifically, we can represent $Ef$ as

$$Ef = \int f(x) \mathbb{P}(x|z) dx = \left\langle \mu(z), \underbrace{\int f(x) \varphi(x) \mathbb{P}(x) dx}_{w_f} \right\rangle. \tag{154}$$

The optimal dual solution to (152) can be represented linearly by $\mu(z)$, *i.e.*,

$$g^*(z) = \mathbb{E}\left[ y - f(x) | z \right] = \underbrace{\mathbb{E}\left[ y | z \right]}_{\langle \mu(z), w_y \rangle} - \langle \mu(z), w_f \rangle = \langle \mu(z), w \rangle.$$

For the primal function $f(\cdot)$, we can always separate as $f(x) = f^{\parallel}(x) + f^{\perp}(x)$, where

$$f^{\parallel}(\cdot) \in \mathcal{K}(\varphi), \quad f^{\perp}(\cdot) \in \mathcal{N}(\varphi), \tag{155}$$

or equivalently,

$$f^{\parallel}(\cdot) = v_{\parallel}^{\top} \varphi(\cdot), \quad f^{\perp}(\cdot) = v_{\perp}^{\top} \varphi^{\perp}(\cdot), \tag{156}$$

where $\varphi(\cdot)$ and $\varphi^{\perp}(\cdot)$ are the basis for $\mathcal{K}(\varphi)$ and $\mathcal{N}(\varphi)$ with the condition $\langle \varphi, \varphi^{\perp} \rangle_{\mathbb{P}(x)} = 0$. Then, we have

$$\mathbb{E}_{x|z}\left[ f(x) \right] = \int \mathbb{P}(x|z) f(x) dx = \left\langle \mu(z), \int \varphi(x) f(x) \mathbb{P}(x) dx \right\rangle \tag{157}$$

$$= \left\langle \mu(z), \int \varphi(x) \left( f^{\parallel}(x) + f^{\perp}(x) \right) \mathbb{P}(x) dx \right\rangle \tag{158}$$

$$= \left\langle \mu(z), \int \varphi(x) f^{\parallel}(x) \mathbb{P}(x) dx \right\rangle + \underbrace{\left\langle \mu(z), \int \varphi(x) f^{\perp}(x) \mathbb{P}(x) dx \right\rangle}_{=0}, \tag{159}$$

which indicates that the effective parametrization of $f(x)$ is $f^{\parallel}(x)$. Therefore, we can also only consider $f^{\parallel}(x) = v^{\top} \varphi(x)$ only, *without any loss*. With these understanding, we have the primal-dual optimization for IV as

$$\min_{v} \max_{w} \; \mathbb{E}_{x,y,z} \left[ w^{\top} \mu(z) \left( y - v^{\top} \varphi(x) \right) - \frac{1}{2} \left( w^{\top} \mu(z) \right)^2 \right] + \lambda \Omega\left( v^{\top} \varphi(x) \right). \tag{160}$$

For other causal inference problems, *e.g.*, IV with observed confounders and PCL, we can also reveal the spectral representation structure. Please refer to (Sun et al., 2024) for details.

### 9.3 Control and Reinforcement Learning

The spectral representation has already been applied in control (Ren et al., 2025), imitation learning (Ma et al., 2025), off-policy evaluation (Hu et al., 2024) and reinforcement learning (RL) (Zhang et al., 2022; Ren et al., 2022b;a; Zhang et al., 2023a; Shribak et al., 2024; Ma et al., 2024; Nabati et al., 2025) for flexible modeling, efficient planning and exploration.

Concretely, given a Markov Decision Process (MDP) specified by a tuple $\mathcal{M} = \langle \mathcal{S}, \mathcal{A}, \mathbb{P}, r, \gamma, d_0 \rangle$, where $\mathcal{S}$ is the state space, $\mathcal{A}$ is the action space, $\mathbb{P} : \mathcal{S} \times \mathcal{A} \to \Delta(\mathcal{S})$ specifies the transition probability of states, $r : \mathcal{S} \times \mathcal{A} \to \mathbb{R}$ is the instantaneous reward function, $\gamma \in [0, 1)$ is the discounting factor and $d_0 \in \Delta(\mathcal{S})$ is the initial state distribution. In both control and RL, we are seeking for the optimal policy $\pi(\cdot|s) : \mathcal{S} \to \Delta(\mathcal{A})$ that maximize the expected cumulative return, defined as:

$$\mathcal{J}(\pi, \mathcal{M}) := \mathbb{E}_{a_t \sim \pi(\cdot|s_t), s_{t+1} \sim \mathbb{P}(\cdot|s_t, a_t)} \left[ \sum_{t=0}^{\infty} \gamma^t r(s_t, a_t) \Big| s_0 \sim d_0 \right]. \tag{161}$$

The major differences between control and RL mainly lies in the information provided: in control setting, the full information of $\mathcal{M}$ is given, while in RL setting, we can only access to $\mathcal{M}$ by sampling.

We further define the state value functions $V^\pi : \mathcal{S} \to \mathbb{R}$ and state-action value functions $Q^\pi : \mathcal{S} \times \mathcal{A} \to \mathbb{R}$ of a given policy $\pi$ as:

$$V^\pi(s) := \mathbb{E}_{a_t \sim \pi(\cdot|s_t), s_{t+1} \sim \mathbb{P}(\cdot|s_t, a_t)} \left[ \sum_{t=0}^{\infty} \gamma^t r(s_t, a_t) \Big| s_0 = s \right],$$

$$Q^\pi(s, a) := \mathbb{E}_{a_t \sim \pi(\cdot|s_t), s_{t+1} \sim \mathbb{P}(\cdot|s_t, a_t)} \left[ \sum_{t=0}^{\infty} \gamma^t r(s_t, a_t) \Big| s_0 = s, a_0 = a \right]. \tag{162}$$

Equivalently, the task can be conveniently defined as finding $\pi^* = \text{argmax}_\pi \ \mathbb{E}_{s \sim d_0} [V^\pi(s)] = \text{argmax}_\pi \ \mathbb{E}_{s \sim d_0, a \sim \pi} [Q^\pi(s, a)]$. Based on the definition of $V^\pi$ and $Q^\pi$, we have the following Bellman recursion holds:

$$Q^\pi(s, a) = r(s, a) + \gamma \mathbb{E}_{s_{t+1} \sim \mathbb{P}(\cdot|s_t, a_t)} [V^\pi(s_{t+1})]$$

$$= r(s, a) + \gamma \mathbb{E}_{s_{t+1} \sim \mathbb{P}(\cdot|s_t, a_t), a_{t+1} \sim \pi(\cdot|s_{t+1})} [Q^\pi(s_{t+1}, a_{t+1})]. \tag{163}$$

We consider the spectral decomposition to $\mathbb{P}(s'|s, a)$, *i.e.*,

$$\mathbb{P}(s'|s, a) = \langle \varphi(s, a), \mu(s') \rangle, \quad \text{and} \quad r(s, a) = \langle \varphi(s, a), \theta_r \rangle. \tag{164}$$

As argued in (Ren et al., 2025), the second condition that representing $r(s, a)$ linearly by $\varphi(s, a)$ is easy to implement by argumenting original spectral representation for $\mathbb{P}(s'|s, a)$ by $r(s, a)$. Then, for any policy $\pi$, we can represent the $Q^\pi$ linearly by $\varphi(s, a)$,

$$Q^\pi(s, a) = r(s, a) + \gamma \int_{\mathcal{S}} \mathbb{P}(s'|s, a) V(s') \mathrm{d}s' = \langle \varphi(s, a), \theta_r \rangle + \gamma \left\langle \varphi(s, a), \int_{\mathcal{S}} \mu(s') V(s') \mathrm{d}s' \right\rangle$$

$$= \left\langle \varphi(s, a), \underbrace{\theta_r + \gamma \int_{\mathcal{S}} \mu(s') V(s') \mathrm{d}s'}_{\boldsymbol{\eta}^\pi} \right\rangle, \tag{165}$$

which implies efficiently temporal difference learning for fitting $Q^\pi$ by least square optimization (Ren et al., 2025), and tractable uncertainty estimation for exploration (Zhang et al., 2022; Ren et al., 2022c).

We only briefly introduce here about exploiting spectral representation for MDP planning. Please refer to (Gao et al., 2025) for more details about (partially observable) MDP learning, planning and exploration with comprehensive comparison.

## 10    Conclusion

After this long journey, we have rigorously revealed the unified key view behind representation learning algorithms, from the class component analysis, to manifold learning, to modern self-supervised learning:

*All of these algorithms are aiming for extracting spectral representation for pairwise mutual information.*

This view not only reveals the long-standing confusion about the sufficiency of representation and organizes the existing algorithms in a clear way for new researchers, but also provides tools for algorithms analysis and design, which inspires a few of novel algorithms as we presented.

However, we emphasize that we indeed observe empirical performance improvement gaps in different algorithms, which implies the **parametrization** and **optimization** for spectral representation matters. For example, the direct spectral decomposition parmatrization (5) induces downstream linear predictor, but with finite-dimension assumption, therefore, limited expressiveness; while the energy-based spectral representation (50) eliminates the finite-dimension assumption and becomes more flexible, but nonlinear downstream predictor (51). Meanwhile, the optimization objectives and algorithms for spectral representation seeking are also important, while the traditional understanding through contrastive vs non-contrastive is incomplete: the major criticism to contrastive SSL is large-batch size, which is not the caused by constrastive loss but the bias from gradient estimator. As we specified in Section 3.1, there can be bias also induced in non-contrastive losses, *e.g.*, VICReg and Barlow-Twins, leading to suboptimal solutions with performances degeneration.

Ablating the effect of parametrization and optimization and finding the optimal empirical configuration is out of the scope of this paper, and we will leave as our future work.

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
