# OpenReview forum: "Spectral Ghost in Representation Learning: from Component Analysis to Self-Supervised Learning"
_TMLR — Withdrawn by Authors_

### Review · Reviewer_rZXa · 2026-03-13

**Summary Of Contributions:**

This paper studies self-supervised learning (SSL) -- i.e., learning some representation of unlabled covariates $x$ that can then be easily built upon to solve various downstream regression tasks of responses $y$ on the covariates $x$ (i.e., learning the distribution $P(y|x)$). The paper notes that there are a lot of approaches to SSL in the literature. However, there have been only a few attempts at unifying these approaches. The paper proposes to use a "spectral" representation for the distribution of $x$, which is sufficient for representing any downstream $P(y|x)$. The paper then looks into how many existing methods fit under the umbrella of this spectral approach.

Key strengths:
- The idea of unifying many disparate methodologies is important and can lead to new insights

Key weaknesses:
- The paper is not well-written to the point that it was not understandable. I've focused pretty much all of my comments below on this point.

**Audience:**

No

**Audience Explanation:**

Given the clarity issues outlined above, I can't exactly tell what the major findings are of the paper. Maybe certain TMLR members would be interested if they could follow the paper, but I'm not sure.

**Broader Impact Concerns:**

No concerns.

**Claims And Evidence:**

No

**Claims Explanation:**

I would in particular say that the claims made are not supported by *clear* evidence.

## Explanation of overall mathematical framework

Section 2 seems to be defining the mathematical framework used to describe various SSL methods throughout the rest of the paper. But after reading Section 2 three times, I really cannot tell what the paper is trying to say. There are just too many pieces of undefined or seemingly contradictory notation, unclear assumptions, and lack of explanation or precise definition for me to follow the development. I've bulleted out the issues that I spotted below:

- "A straightforward idea is seeking the complete space composed by all the subspaces $\phi(x)$ for all $P (y \mid x)$, denoted as $\psi (x)$." I thought that $\phi(x)$ was a function, which seems like a vector, rather than a subspace. E.g., equation (2) is using $\phi(x)$ as an argument to an innner product, which seems like a vector to me. But then $\psi(x)$ seems like a subspace in this sentence. But the notation $\psi(x)$ seems to indicate that it's just a function (a vector). I'm not sure what $\psi(x)$ is supposed to be here as a result. I'm totally lost by the end of this paragraph. Why is the indexing variable switching from $x$ to $s$ to $z$ and then back to $x$? I don't know what the "core set" is being defined as or how it differs from the subspace $\psi$. All of the stuff in here really needs a formal definition environment.
- Immediately after this paragraph is Eq (3), which develops a formula for $P(x'|x)$, i.e., for covariates given other covariates. The paper hasn't previously discussed there even being a distribution over covariates -- it has only introduced the conditional model $P(y|x)$. So what is $P(x'|x)$? Are the covariates not independent of one another here, as they are in many common theoretical setups in machine learning? And why do we care about the distribution of $x' | x$ in the first place? And then later in this equation, we have $x|z$ and $z|x$, and then the marginal over $z$ and the marginal over $x$, none of which were defined (I don't think $z$ was ever even defined). There's also an outer product $v(z)v(z)^T$, but as far as I can tell, $v$ is a function that may or may not be vector-valued (e.g., it seems like it could be discrete-valued) so I'm not sure what this outer product means. Alternatively, earlier, it seemed like $v$ was being used as an approximation to $\psi$, and $\psi$ is a subspace; in this case, the notation $v(z)v(z)^T$ also doesn't make sense. And as a side point, Eq (3) uses $P(z)$ in the denominator -- are we sure that $P(z) > 0$ for all $z$? I didn't see this stated anywhere.
- $A \in R^{d\times d}$ is a full-rank positive-definite matrix" -- so this is assuming that $v$ is a function that maps to $R^d$. Which seems to be implying that we're working over real-valued functions *and* that the feature representation $\phi$ is not infinite dimensional. Both of these need to be stated as assumptions up-front, as this very much limits the types of SSL problems this paper applies to. Finally, how do we know that $A$ is full rank?

The paper then goes on to examine how this spectral framework applies to various existing SSL methods. I tried to read through these sections to get a clearer idea of what the mathematical framework is that the paper is proposing, but I didn't find them helpful. I did notice a few trends in these sections that I think should be corrected.



## Unclear relation to papers being explained

As the paper is trying to explain how its new framework covers pre-existing papers, I think the paper needs to clearly establish what each previous paper is doing and then clearly explain how this fits into the proposed new framework. For example:

- "Follow [sic] the notation from Zbontar et al. (2021)" -- This follows the notation from Zbontar et al. (2021), but Eq (14) from the current paper is not the same final objective as shown in Eq (1) of Zbontar et al. (2021). So I think something is wrong here.
- "which has been investigated in (HaoChen et al., 2021..." A quick look through HaoChen et al. (2021) shows some equations that look similar to Eq (9), but I can't exactly see a match.

As a related "related work" issue, the paper cites Balestriero and LeCun (2022), but doesn't discuss it beyond noting that it is one of "a few attempts to clarify the connections [between SSL methods]." But Balestriero and LeCun (2022) seem to have exactly the same purpose as the current paper -- unifying SSL methods under a common mathematical framework. What are the gaps in this previous work and how is the current paper filling in the gaps?


## Unexplained jumps in logic

"As explained in (Guo et al., 2025), the update (28) is obtained by combining Gram-Schmidt process for orthogonalization to the gradient update of [Eq 29 follows]" -- This is just a random example of a problem that's pretty constant throughout the paper: I'm pretty confident that the relationship between (30) and (28) is not clear to most (all?) readers. If I stare at it for a bit, I can kinda sorta see it. But I have no idea where the Gram-Schmidt orthogonalization comes in. But this asks the point: why is this comment and the subsequent equations here at all? What are they contributing? Does this add significant value to the readers of this paper? Currently, my sense is that the answer to this final question is "no", and this is true of most of the math in the paper. I would recommend to the authors to really think about why each equation and sentence is in the paper and remove it (or, at the least, move it to an appendix) if it's not directly contributing to the purpose of the paper.

## Grammar

There are a lot of grammar issues throughout the paper, even in the abstract. For example, the second sentence of the abstract says "SSL extracts the representation from massive unlabeled data, which will be transferred to a plenty of down streaming tasks."

## Rest of the paper

In the spirit of full disclosure, my comments above are really only from the first 7 pages of the paper. I tried to follow the rest of the paper, but at some point, was just too lost to be able to give any detailed comments, given that I couldn't follow the core methodology development from Section 2. In any case, as far as I could tell, the problems above still hold throughout the rest of the paper.

**Requested Changes:**

I have a few suggestions for the authors, all of which are, in my mind, critical for acceptance:

1. Make sure all notation is clearly defined
2. Make sure all assumptions are clearly stated, preferably in latex "Assumption" environments
3. Break out major results into Proposition / Lemma / Theorem environments, and put extraneous mathematical developments into an Appendix
4. Make sure every equation is explained, and that every development that stays in the main text has a good reason to be there -- think about what the main story of the paper is and whether or not a reader needs to see each equation / statement / figure to be able to grasp that main story.

For examples of doing #1-#4, I would use as inspiration other theoretical papers published in TMLR or similar venues (NeurIPS, ICML, ICLR, AISTATS). For example, just paging through recent TMLR publications, I see this paper: https://openreview.net/pdf?id=gEVPlLhoNI. You can see the difference here -- results are clearly broken out, and the more dense, mathematical portions of the paper are put away in the Appendix.

One measure that really. makes the current paper stand out is how many equations are present in the main text of the paper -- the current paper has 165, which is a huge number! As another example, take HaoChen et al. (2021) (https://arxiv.org/pdf/2106.04156) as cited in the current paper. HaoChen et al. (2021) has 19 pages of main text and, in that time, includes 17 numbered equations plus a handful of non-numbered equations. And HaoChen et al. breaks down their results into many theorem / lemma / proposition / definition environments that makes the paper much more readable. For comparison, the current paper contains 105 numbered equations in its first 19 pages of text, and then the main text goes on for a total of 28 pages with 165 equations. And I think I saw just one proposition environment (no theorems, lemmas, definitions, assumptions, etc.). What I'm trying to get across here is that the current paper is really just not very comprehensible with how much math -- and how little explanation or other structure -- it contains.

---

### Review · Reviewer_WTtv · 2026-04-10

**Summary Of Contributions:**

The paper presents a unifying perspective on representation learning, particularly self-supervised learning (SSL), through the decomposition of pointwise mutual information between samples and their learned representations. It ambitiously covers fundamental SSL techniques, including contrastive, non-contrastive, multi-modal methods, dimensionality reduction, and reinforcement learning.

**Strengths:** The strength of this work is its ambitious scope. By attempting to construct a unified theoretical framework, it connects a wide range of representation learning techniques under a single conceptual umbrella.

**Weaknesses:** The major weaknesses are execution and clarity. The paper requires substantial rewriting to meet publication standards. It suffers from undefined mathematical notations, highly confusing core premises between Singular Value Decomposition (SVD) versus standard scalar product, incomplete writing, and significantly over-claimed contributions that lack the necessary empirical validation.

**Audience:**

Yes

**Audience Explanation:**

The TMLR audience would be interested in the theoretical unification of representation learning. The scope of the paper is highly ambitious, aiming to provide a single framework that encompasses a wide variety of modern techniques, from contrastive and non-contrastive SSL objectives to dimensionality reduction and RL. If the mathematical foundations are rigorously clarified and the claims properly scoped, this holistic perspective would be highly valuable to researchers working on deep learning theory and self-supervised learning.

**Broader Impact Concerns:**

There are no immediate ethical implications noted in the review that would require a Broader Impact Statement. The work is theoretical in nature and focuses on the mathematical properties of representation learning algorithms.

**Claims And Evidence:**

No

**Claims Explanation:**

The claims in the submission are not adequately supported, primarily due to fundamental clarity issues, large gaps between mathematical derivations, and a complete lack of empirical evidence for claims that require it. The clarity issues are severe enough that the paper requires substantial rewriting.

1. **Over-claimed Contributions:** The paper claims to present "a variety of novel scalable representation learning algorithms," but these algorithms are neither explicitly detailed nor empirically implemented. For example, Section 3.2 on "Unnormalized Distribution Fitting" ends abruptly at Equation 40 without explaining how it translates into a usable algorithm. The final two contributions raised in the paper explicitly require experimental validation, yet the paper contains no experiments. Furthermore, claims regarding "characterizing desired properties to the sufficient representation" remain unclear in Section 2.
2. **Mathematical Clarity and Core Premises:** The core premise and notation for the SVD (Section 2) are highly confusing. The paper uses standard scalar product notation ( $\langle \cdot , \cdot \rangle$ ) to define the SVD of $\mathbb{P}(y∣x)$ , without explaining how SVD can be defined directly as a scalar product. Then, the same notation is used again to denote the standard scalar product. This casts doubt on the validity of the "spectral" terminology used throughout the text. Additionally, critical derivations are missing or unclear (e.g., the transition from Eq. 5 to 6 mentioned an unnecessary multiplication with $\tilde{\mathbb{P}}(x)$ , the link between Eq. 10 and 11, or the derivation of Eq. 17 using the "penalty method").
3. **Vague SSL Remarks:** Observations regarding specific SSL algorithms are insufficiently supported. For instance, the paper claims certain gradient calculations in Barlow-Twins are biased without explicitly showing the gradient, it mentions terms in VICReg that are incompatible with SGD without specifying them, or it states BYOL can easily yield an unbiased stochastic gradient estimator without providing the mathematical derivation.

**Requested Changes:**

**Critical for acceptance:**

1. **Substantial Rewriting for Clarity:** The paper requires thorough rewriting to improve readability and correct writing errors. This includes fixing unfinished sentences (e.g., in the Abstract and Conclusion), awkward phrasing, and "Proof" sections that are not linked to a formal proposition or theorem (in page 11). These issues are frequent enough to distract the reader.
2. **Clarify Notation and Derivations:** Ensure all notations are properly introduced before use. For instance, clarify the difference between $\mathbb{P}$ and $\tilde{\mathbb{P}}$ , define $p_{\theta}$ and $\theta$  for Eq. 59, and clearly introduce $w$ and $u$ before Eq. 55. Add more details about missing derivation steps, e.g. Eqs. 6, 11, and 17.
3. **Resolve SVD / Scalar Product Confusion:** Revise Section 2 to clearly distinguish between SVD and standard scalar products. Ensure the scalar product notation ( $\langle \cdot, \cdot \rangle$ ) is properly justified to represent SVD, which would support the "spectral" framing of the paper.
4. **Provide Empirical Evidence or Adjust Claims:** Either provide experimental validation for the "novel scalable representation learning algorithms" and the last two contributions, or significantly tone down the claims to match a purely theoretical/conceptual paper.
5. **Substantiate Claims in Remarks:** Provide the exact mathematical terms and gradients when making claims in the remarks, e.g., about Barlow-Twins (biased gradients), VICReg (SGD incompatibility), or BYOL (unbiased estimator).

---

### Review · Reviewer_D8c6 · 2026-04-10

**Summary Of Contributions:**

**Summary of Contributions**

This paper proposes a unified spectral framework for self-supervised learning (SSL). The central thesis is that the SVD of the conditional operator P(y|x) defines a "spectral representation" φ(x) that is sufficient for downstream prediction, and that this spectral structure is the common target implicitly shared by a wide variety of SSL algorithms, contrastive, non-contrastive, energy-based, and latent-variable methods alike.
The paper proceeds as follows: Section 2 establishes the sufficiency of the spectral representation and argues it can be recovered from unlabeled data via the spectral decomposition of pointwise mutual information (PMI). Sections 3–6 categorize existing SSL methods (direct spectral, energy-based, latent-variable, nonlinear) as different parameterizations of this common spectral target, and propose new algorithms within each family. Section 7 extends the framework to multi-modal settings. Section 8 connects SSL to classical component analysis methods (PCA, CCA). Applications to controllable generation, reinforcement learning, and causal inference are also discussed.

---


**Strengths**

1. **The paper provides an interesting angle of looking into learning mechanics of self-supervised learning:** The formulation nicely dissociates labelled task objective to learning SVD of the joint distribution of neighbourhood samples/augmentations and the task specific weight matrix.

2. **The paper presents an ambitious attempt to bring contemporary SSL techniques into a single umbrella:**
- The paper covers SimCLR, MoCo, BYOL, SimSiam, Barlow Twins, VICReg, CLIP, SigLIP, DINO, SwAV, DeepCluster, SeLa, Word2Vec, diffusion-based representations, and connects them all to PCA, CCA, MDS, Isomap, Laplacian Eigenmaps, t-SNE, and Locality Preserving Projections. I'm not aware of another single paper that provides formal connections across all of these.
Some of the observations that come out of this exercise are interesting: like the requirement of non-linear probes for kernel based objectives in SimCLR; The energy-based to attention mechanism connection (Eq 56) is novel and elegant.
-  The paper identifies a real theoretical gap in the field.
The paper identifies that the field has been operating without clear answers to: "what makes a representation sufficient?" and "why do so many different methods work?" These are the right questions, and framing them through the lens of spectral theory is a natural and principled approach.

3. **The historical connections (Section 8) provide genuine perspective.** Showing that PCA, CCA, MDS, Isomap, and Laplacian Eigenmaps are all special cases of the same spectral representation framework that modern SSL methods optimize, with specific parameterization choices made explicit, is interesting and loops back to these fundamental methods.

---
**Negatives**


1. **Presentation of the paper:**

This is the biggest problem with this work. The paper is difficult to understand in its current form. The derivations provided throughout are quite dense, and oftentimes abruptly expanded or reduced without proper citation to prior assumptions or equation numbers. It took me a long time to verify the equations, and I often struggled with keeping track of how the equations were evolving. There is a serious lack of structure to the paper. While the paper touches upon a great diversity of prior SSL methods, it has been done quite hastily. A lot of derivation could have been expanded in the supplementary and the main idea could have been provided in the main text. Also, the derivations that are in the main text, skip over steps or often lack referencing prior equations which makes it difficult to follow.

 2. **The P(x'|x) formulation conflates data structure with augmentation strategy.**

Throughout the paper, P(x'|x) is treated as a property of the data distribution. In practice, it's entirely determined by the data augmentation strategy (crop, color jitter, etc.). The paper never discusses how the choice of augmentations affects which spectral components are recoverable, whether misspecified augmentations lead to incomplete representations, or how the framework guides augmentation design. This is a significant gap between theory and practice.

3. **The Universality Argument via P(x'|x) Has Unexamined Assumptions**

Section 2 addresses the task-dependence problem: since different tasks P(y|x) yield different φ(x), how can SSL provide a universal representation? The paper's solution is elegant in structure, define a "complete space" φ(x) spanning all task-specific subspaces, then show this can be recovered from the PMI of P(x, x')/[P(x)P(x')] without labels. However, several issues remain:

(a) Dimensionality of the complete space. The complete space φ(x) is the union of all task-specific subspaces across all possible P(y|x). The paper requires φ(x) to be full-rank but never bounds its dimension. The finite-d approximation concern from above becomes strictly worse here, the complete space is at least as large as any individual task's subspace, and potentially much larger.

(b) Dependence on augmentation structure. The PMI-based recovery of φ(x) relies on P(x'|x), which in practice is determined by the data augmentation strategy (e.g., cropped/color-jittered views of the same image). Whether this P(x'|x) actually spans all task-relevant subspaces depends entirely on the augmentation choice. If augmentations are too narrow, the resulting representation misses task-relevant structure; if too broad, it includes noise. This augmentation-task alignment issue is not discussed.

(c) Conditional independence assumption. The derivation of Equation 3 integrates P(x'|z)P(z|x) over z, which requires the core variable z to make x and x' conditionally independent. This is a strong structural assumption about the data generating process that is not stated or justified.

4. **Not sure if equation 13 and 14 will share the same solution: Eq 13–14 equivalence assumes realizability.**

 The paper claims that the spectral objective (Eq 13, with λ=1) and the Barlow Twins objective (Eq 14, with arbitrary λ) "share the same solution." This holds only when the model has sufficient capacity to achieve the global optimum C = I, at which point both loss terms vanish simultaneously and the weighting λ becomes irrelevant. However, when C = I is not achievable, as is typical with finite-capacity networks, the two objectives induce different tradeoffs between variance preservation (diagonal terms) and decorrelation (off-diagonal terms), leading to genuinely different solutions for different λ. Notably, the original Barlow Twins paper (Zbontar et al., 2021) reports strong sensitivity to the choice of λ, with λ ≈ 0.005 substantially outperforming λ = 1. This directly suggests that the realizability condition does not hold in practice, and the claimed equivalence is only valid at the population level with infinite model capacity. The paper should state this assumption explicitly and discuss the implications for finite-capacity models, where the spectral framework's predictions diverge from observed empirical behavior.

5. **Does the batch size hypothesis hold in experiments?**

- The results from BYOL paper shows, BYOL also has worse performance with lower batch size
- The Bartlow twins paper shows it is more robust to lower batch sizes than BYOL.
- At what batch size would BYOL start degrading?

6. **No Empirical Validation:**

The paper is entirely theoretical, however provides claims about unifying and explaining the success of practical SSL algorithms. No experiments are provided to demonstrate that: (a) the proposed new algorithms (Sections 3.2, 4.2, 5.2) work in practice; (b) the spectral view provides actionable insights, e.g., the spectral gap predicts downstream performance; or (c) the quality of different methods' approximation to the spectral target correlates with their empirical performance.

7. **Lack of comparison or insufficient attribution to prior theoretical work in SSL:**
 There has been significant progress on the theory of SSL, but the paper lacks sufficient comparison against them. A discussion of what new ideas or insights this work offers compared to existing theoretical frameworks is missing.

- The paper's core theoretical contribution, that SSL methods target the spectral decomposition of P(x,x')/[P(x)P(x')], and that this spectral representation is sufficient for downstream tasks, has significant overlap with Johnson et al. (ICLR 2023, 'Contrastive Learning Can Find An Optimal Basis For Approximately View-Invariant Functions'), which is not cited. Johnson et al. establish the same PMI/positive-pair kernel as the common target, prove minimax optimality of the eigenfunction representation for downstream linear prediction, and provide finite-dimensional generalization bounds.

- Several other highly relevant theoretical works are also not cited or discussed. For instance, Arora et al. (ICML 2019) provide provable downstream guarantees and sample complexity bounds. Tosh et al. (ALT 2021) prove near-optimality of linear functions of contrastive representations under a multi-view redundancy condition that relies on the same conditional independence assumption used implicitly in Eq. 3, and provide explicit dimensionality bounds addressing the finite-d truncation. Wang & Isola (ICML 2020), one of the seminal theoretical SSL papers, decompose the contrastive loss into alignment and uniformity terms that correspond to the positive-pair and negative-pair terms in Eq. 9 etc, like: Zimmermann et al. (ICML 2021), Zhuo et al. (ICLR 2023) etc. The paper should position itself explicitly relative to these works and discuss them w.r.t. the proposed work. (Please see the [Ref])


8. **Lack of explanation for design choices of SSL:**

- Many of the SSL methods like SimSiam, BYOL etc for instance depend on strict design choices like Stop-gradient, high learning rate of predictor for non-collapse solution. They have not been shown.
- Another example: clustering based methods like SwAV can learn meaningful (good) features without training the clustering and fixing it at the beginning.

---

**[Ref]**

Johnson, D. D., Hanchi, A., and Maron, H. Contrastive Learning Can Find An Optimal Basis For Approximately View-Invariant Functions. ICLR, 2023.

Arora, S., Khandeparkar, H., Khodak, M., Plevrakis, O., and Saunshi, N. A Theoretical Analysis of Contrastive Unsupervised Representation Learning. ICML, 2019.

Tosh, C., Krishnamurthy, A., and Hsu, D. Contrastive Learning, Multi-View Redundancy, and Linear Models. ALT, 2021.
Wang, T. and Isola, P. Understanding Contrastive Representation Learning through Alignment and Uniformity on the Hypersphere. ICML, 2020.

Zimmermann, R. S., Sharma, Y., Schneider, S., Bethge, M., and Brendel, W. Contrastive Learning Inverts the Data Generating Process. ICML, 2021.

Zhuo, Z., Wang, Y., Ma, J., and Wang, Y. Towards a Unified Theoretical Understanding of Non-Contrastive Learning via Rank Differential Mechanism. ICLR, 2023.


---

**Minor:**

- <.,.> is a scalar product (?), hence the use of this notation for eq 10, with <scalar, matrix> may not be correct.

- Probably there should be a transpose on the φ(x) in the first term under<> in eq 10.

- Second and third terms on the RHS of the equation prior to eq18 should not be Identity, they should be scalar?

**Audience:**

Yes

**Audience Explanation:**

The paper explores an important question about bridging the gap intrinsic to otherwise empirical methods in SSL. The study can be helpful to design more principled and theoretically grounded methods in SSL. I think this paper is useful for the representation learning community.

**Claims And Evidence:**

No

**Claims Explanation:**

- The presentation of the paper requires serious improvement. (See weakness 1).
- The paper also lacks proper discussion on design choices these SSL methods use that are critical for non-collapse solutions.
- The paper lacks empirical justifications of the predictions made.

**Requested Changes:**

Please see the major weakness comment.

---

### Note · Authors · 2026-04-10

I have read and agree with the venue's withdrawal policy on behalf of myself and my co-authors.